# Dynamic hydrological discharge modelling for coupled climate model simulations of the last glacial cycle: The MPI-DynamicHD Model version 3.0

Thomas Riddick[1], Victor Brovkin[1], Stefan Hagemann[1,*], and Uwe Mikolajewicz[1]

[1]Max Planck Institute for Meteorology, Bundesstraße 53, 20146 Hamburg, Germany
[*]*Now at:* Institute of Coastal Research, Helmholtz-Zentrum Geesthacht, Max-Planck-Straße 1, 21502 Geesthacht, Germany
*Correspondence to:* Thomas Riddick (thomas.riddick@mpimet.mpg.de)

**Abstract.** The continually evolving large ice sheets present in the Northern Hemisphere during the last glacial cycle caused significant changes to river pathways both through directly blocking rivers and through glacial isostatic adjustment. Studies have shown these river pathway changes had a significant impact on the ocean circulation through changing the pattern of freshwater discharge into the oceans. A coupled Earth System Model (ESM) simulation of the last glacial cycle thus requires a hydrological discharge model that uses a set of river pathways that evolve with Earth's changing orography while being able to reproduce the known present-day river network given the present-day orography. Here we present a method for dynamically modelling river pathways that meets such requirements by applying pre-defined corrections to an evolving fine scale orography (accounting for the changing ice sheets and isostatic rebound) each time the river directions are recalculated. The corrected orography thus produced is then used to create a set of fine scale river pathways and these are then upscaled to a coarser scale on which an existing present-day hydrological discharge model within the JSBACH land surface model simulates the river flow. Tests show that this procedure reproduces the known present-day river network to a sufficient degree of accuracy and is able to simulate plausible paleo river networks. It has also been shown this procedure can be run successfully multiple times as part of a transient coupled climate model simulation.

## 1  Introduction

Results of ocean circulation models are very sensitive to freshwater flux (Maier-Reimer and Mikolajewicz, 1989; Schiller et al., 1997; Stouffer et al., 2006; IPCC, 2013). The accurate modelling of ocean circulation requires the river run-off to be correct for individual ocean basins and distributed with a roughly accurate spatial pattern around each basin's edge. During the last glacial cycle, the courses of rivers in North America, northern Europe and Siberia were significantly altered by a combination of the physical presence of the ice sheets directly blocking the flow of rivers and the effects of isostatic adjustments altering the orography of ice free areas (Teller, 1990; Licciardi et al., 1999; Mangerud et al., 2004; Wickert, 2016). Previous studies indicate that modelling of these alterations may play an important role in the success of a transient simulation of the last glacial cycle (Alkama et al., 2008). A comparison of a reconstructed orography for the last glacial maximum (LGM) to a present day orography indicates that the most significant changes in orography occurred close to the ice sheet. Africa, much of South

America and southern Asia were only weakly affected by the changes in the orography. Here we introduce a dynamical model of river pathways and hydrological discharge for the simulation of glacial cycles that accounts both for the physical presence of ice sheets and for isostatic adjustments.

JSBACH (Reick et al., 2013; Schneck et al., 2013) is the land surface scheme of the Max Planck Institute for Meteorology's Earth System Model MPI-ESM (Giorgetta et al., 2013). In this paper we use JSBACH 3.0, which has undergone further developments since the version (JSBACH 2.0) used for the Coupled Model Intercomparison Project 5 (Taylor et al., 2012) as described in Giorgetta et al. (2013). These developments are a new soil carbon model (Goll et al., 2015) and a new five layer soil hydrology scheme (Hagemann and Stacke, 2015) instead of the previous bucket scheme.

In JSBACH lateral freshwater fluxes are treated by the Hydrological Discharge (HD) model (Hagemann and Dümenil, 1998b; Hagemann and Dümenil Gates, 2001). Although the HD model is included in JSBACH it can also be run independently as a standalone model. In this model lateral freshwater fluxes are split into three components: base flow, overland flow and river flow. Base flow represents the slow movement of water in the lowest layer of the soil, overland flow surface flow outside channels and river flow channelled surface flow. The HD model is run on a $0.5°$ regular latitude-longitude grid with a daily time-step. All three components of the flow from a cell are directed to one of the cells eight direct neighbours. Within each grid cell river flow is modelled through a cascade of linear reservoirs; a cascade of linear reservoirs in each cell is necessary to accurately simulate both the translation characteristics (which determine how fast water passes through the cell) and retention characteristics (which determine how much water is stored in the cell) of each cell. The number of reservoirs $n_r$ is set to $5$ (with the exception of cells containing major lakes; however, such lakes are switched off entirely in the version of the HD model used for dynamic hydrological discharge modelling by this paper as their formulation is unsuitable for modelling lakes that evolve with a changing orography). The river outflow as a function of time $Q(t)$ from each reservoir is modelled as:

$$Q(t) = \frac{S(t)}{k} \tag{1}$$

where $S(t)$ is the water content of the reservoir as a function of time and $k$ is the water retention time (also called the retention coefficient) of each reservoir. The retention time for rivers $k_r$ is calculated (in days) for each cell in the grid thus:

$$k_r = 0.992 \frac{\text{days}}{\text{m}} \cdot \frac{\Delta x}{s^{0.1}}. \tag{2}$$

where $\Delta x$ is the distance to the centre of the next downstream cell from the centre of the cell under consideration (in metres) and defining slope $s = \frac{\Delta h}{\Delta x}$ with $\Delta h$ as the change in orography between this cell and the next downstream cell (in metres). The sign of $\Delta h$ is defined such that $s$ is positive for a downhill slope. $s$ is set to a constant value of 0.00001315 when its original value is either negative or zero. In this paper the set of reservoir retention coefficients for all three of the components of the flow for the whole globe are known collectively as the flow parameters.

In the standard version of the HD model for the present day that is part of JSBACH the direction of flow is decided by a set of manually corrected present day river directions referred to in this paper as the manually corrected (present day) HD model river directions. These manually corrected (present day) river directions are derived by first applying a downslope routing to a pit filled orography; then correcting by hand to ensure the correct paths for the world's major rivers; and finally further correcting by hand the catchments of major rivers based on careful comparison with reference catchments.

The surface run-off and soil drainage of a cell from the JSBACH model are added to the overland flow and base flow respectively and the flow of these through the cell is modelled in each case by a single linear reservoir. The water retention time for overland flow reservoirs is calculated using the average slope within a grid-box itself when considered on a finer scale (the inner-slope) along with the $\Delta x$ as defined above; see Hagemann and Dümenil (1998a) for details. The method for calculating the water retention times for base flow is similar to that given in Hagemann and Dümenil (1998a) but takes into account some spatial variability (Müller, 1998). Base flow retention times tend to be roughly three orders of magnitude longer than those of overland and river flow. The outflow from all three components is summed and this is used as the input for the river flow of the next downstream cell. When evaluated in an inter-model comparison study (Haddeland et al., 2011), the performance of the HD model as a component of the Max Planck Institute Hydrology Model (MPI-HM) (Stacke and Hagemann, 2012) did not differ significantly from those of similar components of other global hydrology models.

The challenge for a paleoclimate simulation is to develop a method for periodically updating the river directions and flow parameters used with sufficient accuracy (Alkama et al., 2008). If the simulation calculates changes in the orography from the output of an interactive ice-sheet model within a wider ESM then another requirement is the river directions and flow parameters can be recalculated quickly when it is necessary to update them. Given the large inaccuracy in the distribution of precipitation likely to occur in paleoclimate simulations it will suffice to capture only the main features of the river directions and the flow parameters especially need only be a rough approximation. The fine details of catchment boundaries and outflow points and the exact temporal response of the discharge model to precipitation events will not be required. Generating river paths is however challenging as the natural scale determining the path of rivers is often far smaller than the scale it is feasible to provide an orography on. Examination by eye of orography datasets shows that narrower river valleys of major rivers are not resolved on a $0.5°$ grid, partially resolved on a 10-minute grid and well resolved on a 1-minute grid (although mistakes in the paths of major rivers, e.g. the Mekong, still occur even if a 1-minute resolution orography is used). Another challenge of generating river directions is false sinks, closed depressions that are artefacts of the Digital Elevation Model (DEM) and don't physically exist. Lack of detail in the orography can mean the height of riverbeds are overestimated at some points in narrow valleys thus leading to apparently closed pits or sinks being found in the orography. False sinks also appear at higher resolutions due to various imperfections in the measurement of orography by satellite (Yamazaki et al., 2017). If river directions are generated from an unmodified orography by the line of steepest descent, then these will be marked as inland sink points, while they are actually unimpeded rivers. Therefore an algorithm is required to either fill-in these false sink points or to let rivers "carve" out of them.

Most previous ESM based simulations of the last glacial cycle have used the technique of extending present day river directions to the sea (e.g. Ziemen et al., 2014). This was a suggested method for the Paleoclimate Modelling Intercomparison Project Phase 3 (PMIP-3) (Braconnot et al., 2011, 2012). A number of authors have tackled the problem of modelling river routing during the last glacial cycle. Wickert (2016) provides river maps for various time points during the deglaciation derived directly from a 30-second orography combined with various ice-sheet reconstructions (alongside a useful comparison of these river maps to known data). However, this technique would be too computationally expensive to run fully automatically every 10 years during a transient simulation. Tarasov and Peltier (2006) present a dynamic river routing and lake model for North

America during the Younger Dryas that is in many ways similar to that presented here and from which the basic principle of upscaling of effective hydrological heights was taken. However, our new model uses a different combination of upscaling techniques and orography corrections from those of Tarasov and Peltier (2006) as well as a different grid.

Most previous simulations of the last glacial cycle that use coupled Global Circulation Models (GCMs) have only treated time-slices; transient simulations having usually been run only in Earth System Models of Intermediate Complexity (EMICs). Goelzer et al. (2012) present a dynamic river routing module for much of the Northern Hemisphere to produce freshwater inflows from ice sheet meltwater (direct precipitation was not considered) for the CLIO ocean model (Goosse and Fichefet, 1999), a component of the LOVECLIM EMIC (Goosse et al., 2010), driven by the ice sheet model NHISM (Zweck and Huybrechts, 2005). There method is to transform the HYDRO1k hydrologically condition present day orography (USGS, 2017) to a 25 km polar stereographic grid by 2-dimensional Langrange polynomials and use this as a base orography to which to add ice sheet height corrections and isostatic corrections to during a simulation of the last deglaciation. They note the need to apply some manual corrections to resolve blocked valleys in the present day orography. The ESM model of Ziemen et al. (2018), which is a precursor to the model the method presented here will be used in, used a simplified method to treat river routing following similar ideas to those presented here.

The first transient synchronously coupled GCM simulation of the deglaciation was Liu et al. (2009). This used a time-varying prescribed forcing to simulate the release of glacial meltwater from rivers. However, the PalMod project (Latif et al., 2016), which the approach presented here is intended for, aims to run simulations that limit external forcings to just solar and volcanic forcings, thus running transient models using a fully self-consistent ESM and clearly precluding a proscribed-forcing based approach to meltwater runoff.

An important test of any method is the ability to accurately generate present-day river directions. Large rivers away from the ice-covered regions of Earth do not appear to have drastically changed their course during the last glacial cycle, so for large areas of Earth the river directions should be the same as the present day river directions for the entire glacial cycle. Early testing showed that generating river directions on a $0.5°$ grid by simply following the line of steepest descent gives unsatisfactory results for present day river directions. Using the same technique on a 10-minute grid gives better results although there are still some mistakes. Our method aims to correct those mistakes such that the present day river directions can be accurately reproduced on a $0.5°$ grid.

The HD model within JSBACH was originally designed for use in near present day simulations with static river directions and flow parameters. Several other components of the MPI-ESM that may also be expected to change on paleo-timescales were likewise designed for use with static input data for near present day simulations, e.g. the land-sea mask. Adapting MPI-ESM to allow the static input data of such components be replaced with a time varying field within the model would in some cases have a considerable negative impact on the model's performance and/or be very technically challenging. (Both of these difficulties would apply in the case of river directions and flow parameters.) As these components only vary comparatively slowly with time compared to the model's time-step it will instead be sufficient to only update them at a set interval of 10 years. Every 10 years during a long transient run MPI-ESM will halt and a number of processing scripts will be run to update otherwise static input data before MPI-ESM is restarted for the next 10 year section with the new input data. The input data updated will

include the river directions and flow parameters. These will then remain constant during the next 10 year section. For transient runs where MPI-ESM is coupled to an ice-sheet model, during each of these decennial updates isostatic corrections will first be calculated using a viscoelastic earth model, for example, the Viscoelastic Lithosphere and Mantel Model (VILMA) (Martinec, 2000) and alongside the height of the ice-sheet be applied to a present day base orography to create a general orography for a given time. This will then act as an input to the process of updating the river directions and flow parameters.

## 2 Method

### 2.1 Overview of method

The starting point for a simulation of the last glacial cycle is a simulated 10-minute resolution orography for time $t$ in the simulation including the height of any ice-sheets and isostatic corrections along with the same 10-minute resolution orography for the present day; this latter orography is hereinafter referred to as the present day base orography. From this pair of orographies the height anomalies for time $t$ with respect to the present can be calculated and applied to a present day reference orography to which we also apply height corrections (as described below); this is necessary as different present day orographies can differ markedly due to differences in their methods of fabrication and thus height corrections must be applied to the particular present day orography they were created for. (Note it would also be possible to work with a pair of orographies on a different resolution then remap the anomalies between them to a 10-minute resolution.)

River directions are regenerated every 10 years by a four step process. A brief outline is given here; detailed descriptions of each step are given in the subsequent sections. A flow diagram outlining the steps is given in Fig. 1. Firstly, the orography for time $t$ is adjusted by subtracting the matching present day base orography and adding a given reference present day orography (see Sect. 2.2 and also the discussion in the preceding paragraph). Secondly, a pre-generated set of relative height corrections for a small number of cells (or all cells in the case of North America where a different algorithm is used to generate the corrections) are added to this orography (see Sect. 2.3). These relative height corrections are such that when applied to the given reference present day orography they return a set of river directions with all of the major errors in river paths and catchments corrected. Thirdly, a set of sink-less river directions on the 10-minute grid is generated using the river carving method of Metz et al. (2011) (see Sect. 2.4). Fourthly, these river directions are upscaled to the $0.5°$ grid required by the HD model (see Sect. 2.5). Flow parameters are generated on the $0.5°$ grid using an upscaled and sink filled version of the 10-minute orography for time $t$ (see Sect. 2.6).

The process described above for the generation of river directions and flow parameters is entirely automatic. Prior to the first application of this process it was necessary to develop the above mentioned pre-generated set of relative of height corrections. This development was guided and evaluated by hand although making extensive use of automated tools to expedite the development process and improve the accuracy of the corrections in certain regions. Alongside these automated tools some corrections were also made by hand. The development of these corrections is discussed extensively in Sect. 2.3.

A useful diagnostic derived from sets of river directions is the total cumulative flow. For each cell this is the total number of upstream cells that flow, directly (i.e. without passing through any other cells first) or indirectly (i.e. passing through other

cells first), into that cell. In this paper we also count the cell itself within the total cumulative flow; thus the total cumulative flow of any cell is equal to the sum of the total cumulative flows of all cells that directly flow into it plus one. Total cumulative flow is a property of the river directions as a dry system and doesn't account for variations in rainfall. It also doesn't account for the variation of latitude-longitude cell surface areas with latitude.

## 2.2 Changing the present day base orography

The first step is to change the present day base orography underlying any given input orography for time $t$ to match the present day reference orography used to generate the DEM corrections. In the case of all the data generated for this paper this present day reference orography was ICE-5G version 1.2 (Peltier, 2004). This is done by applying the following modification to an input orography on a cell-wise basis:

$$h_{\text{working orography}} = h_{\text{standard past orography}} - h_{\text{present day base orography}} + h_{\text{present day reference orography}} \tag{3}$$

where $h_{\text{working orography}}$ and $h_{\text{standard past orography}}$ are the height value of a cell at a given time $t$ in a paleoclimate simulation while $h_{\text{present day base orography}}$ and $h_{\text{present day reference orography}}$ are the present day height value of the given cell from two different DEMs. All these orographies will have a 10-minute resolution. The present day base orography is the base orography that is used by a viscoelastic earth model to produce general purpose orographies for times in the past for a wider ESM (in a setup with a coupled ice-sheet; otherwise, it is the base orography that was used to derive the ice-sheet reconstruction being used). The standard past orography is then the orography derived from the present day base orography (again, in a setup with a coupled ice-sheet; otherwise, it is just the orography reconstruction being used) for time $t$ via glacial height adjustments from the ice-sheet model and isostatic adjustments from the viscoelastic earth model. For any given cell:

$$h_{\text{standard past orography}} = h_{\text{present day base orography}} + h_{\text{glacier}} + \Delta h_{\text{isostatic}} \tag{4}$$

where $h_{\text{glacier}}$ is the vertical thickness of the ice-sheet in the cell (which is set to zero if the cell doesn't contain an ice-sheet) for time $t$ and $\Delta h_{\text{isostatic}}$ is the isostatic adjustment for the cell for time $t$. The present day reference orography is the present day orography used in the derivation of the corrections used in the second step of our method. The working orography is a new intermediary working orography (for time $t$) specifically used by our river direction generation process; it is this the DEM corrections are applied to in the second step of that process.

This first step is necessary because comparisons show due to differing methods of fabrication different 10-minute present day base orographies used in paleoclimate simulations can differ in many cells quite widely (by as much as several hundred metres in height in areas of very high inter-cell height variance). These differences are systematic and are apparently due to biases in the processing of original satellite data (often on a finer scale) to produce the 10-minute present day orography. In the ESM setups this method is intended for the present day base orography will often not be the same as the present day reference orography as the present day base orography is likely to be set for the wider ESM setup the HD model is embedded in while the present day reference orography that all the DEM corrections discussed in the next section were generated for is ICE-5G and it would require significant effort to regenerate these for another reference orography. Preliminary testing showed that the DEM

corrections applied in the second step of our method are only valid for orographies generated from the same present day base orography as the corrections themselves were derived for (i.e. the present day reference orography) and thus it is necessary to adjust the input orography such that these corrections are still valid.

Although it is intended to use this method with all the input orographies on a 10-minute resolution, this step provides the option of alternatively using a paleo-orography and corresponding present day base orography that are of a lower resolution than 10-minutes.

## 2.3 DEM corrections

The set of relative height corrections applied in the second step were derived through comparison with a variety of sources of information on present-day river paths and catchments. The development of these relative height corrections was a onetime task which was guided and overseen by hand even when some elements were automated. This set of relative height corrections is provided as input data to each application of the main river routing and parameter generation process which is in itself fully automatic.

Three techniques were used to generate these corrections: application by hand to individual orography grid points, intelligent river burning applied to selected regions, and upscaling effective hydrological heights from a very high resolution orography. Each of these techniques is described (including definitions of the terms 'intelligent river burning' and 'effective hydrological heights') individually below. In North America upscaling effective hydrological heights was used in combination with the other two methods. Only hand application of corrections and intelligent river burning were applied to the rest of the globe. Both combinations of techniques are expected to produce satisfactory results; the combination of upscaling effective hydrological heights and the other two techniques is expected to produce slightly more accurate results in regions where river pathways changed significantly during the last glacial cycle than just using the other two techniques. However upscaling effective hydrological heights was developed after the other two techniques and it was noted that significant additional effort was required to apply it to North America (justification of this choice of trial region is given below) thus it was decided not to apply it to the rest of the globe.

The application of both of the first two methods was directed by comparison with the manually corrected present day HD model river directions (through plots of total cumulative flow and catchment maps) and by comparing with river directions generated from a finer 1-minute orography (through plots of the total cumulative flow only). Differences were resolved using the catchment data of the HydroSHEDS database (Lehner and Grill, 2013). (Online geographical information from a wide range of sources was used to aid interpretation.) The relative corrections applied by hand usually correct the height of the cells of the 10-minute present day orography to the height of the valley floor of the river under consideration observed in the 1-minute orography. Occasionally some guesswork and judgement had to be applied to decide what the true height of the valley floor was; in a few specific cases the valley was also poorly defined at a 1-minute resolution (e.g. the Iron Gates gorge on the Danube).

Intelligent burning of small manually selected regions (usually short sections of an individual river valley) produces similar results but automates the procedure. River directions are generated for the present day from a "super-fine" 1-minute orography

using the same carving algorithm as described below and the total cumulative flow is generated from these super fine river directions. The 1-minute orography is masked outside a selected region and then further masked within that region where the super fine total cumulative flow is below a given threshold. Then the height of each cell in the 10-minute orography is replaced with the highest unmasked height (if any) within the area of the 1-minute orography that corresponds to that 10-minute cell (as long as that height is lower than the present height of the cell in the 10-minute orography; otherwise it is left unmodified). This quickly burns a river from the super fine 1-minute orography into the 10-minute orography but, unlike regular stream burning techniques (Maidment, 1996; Mizgallewicz and Maidment, 1996; Saunders, 1999), only to the depth observed in a finer orography. Thus the height of the riverine cells in the burnt area remains realistic to within the accuracy of a finer orography thus the possibility of the river changing direction during the glacial cycle due to changes in the orography remains unimpeded. Note stream burning should not be confused with the largely unrelated technique of river carving. The total cumulative flow threshold mentioned can differ for each region where intelligent burning is applied and is set by hand for each case such that only the cells of the super fine orography through which the main river flows in the region of application in question remain unmasked. The results of each application of intelligent burning were examined carefully by eye before proceeding. Once the burning process is complete the changes in the 10-minute orography for the present day are converted to relative changes in height (suitable for application at any time during the glacial cycle) by subtracting the original unmodified version of the orography.

Corrections generated using either one or the other of the first two methods (or a combination of the two) were applied all across the globe to eliminate all significant errors seen in the river directions derived from a 10-minute orography with the exception of some problems related to true sinks which were ignored as true sinks will not be used when generating dynamic river directions. Hand application of corrections was usually used where only a few cells needed to be changed; intelligent burning was used where an error in the river directions for a particular section of a river needed a larger number of corrections to resolve. Similar results could have been achieved using application of corrections by hand alone but this would have been significantly more time consuming. Figure 2 shows how applying an appropriate correction corrects a problem in the catchment of the Danube.

We define the effective hydrological height of a cell within a DEM as the elevation of the river sill within the cell. This is namely the height of the highest point in the 'most likely' river pathway through the cell when examining the internal structure of the cell's height within a much finer DEM. The most likely river pathway is defined as the path whose maximum elevation while transversing the cell is lowest disregarding any paths that don't cross a significant fraction of the cell. The basic principle of upscaling effective hydrological heights was adopted from Tarasov and Peltier (2006). The actual algorithm used here is different from that of Tarasov and Peltier (2006) and was developed specifically for this paper; however, the results achieved should be very similar for most grid cells (differences may occur along catchment boundaries and for narrow channels crossing the border between two cells at a very acute angle). The technique of upscaling effective hydrological heights was only used for North America (the land boundary of its application being the narrowest point of Central America to minimise any possible edge effects) as this was seen as a critical region for paleohydrology and Tarasov and Peltier have previously shown this

technique to be effective in this region. Upscaling effective hydrological heights could also potentially be applied beneficially to Eurasia however we decided against doing so because of the significant additional effort required.

We give here a brief outline of the algorithm; more detailed descriptions of the algorithm are given in Appendices A and B. The algorithm used here works by exploring possible paths through each of a set of sections of a fine orography that correspond to the individual cells of a coarse orography. This is performed by flooding each coarse cell on a fine cell by fine cell basis according to height while filling in any false sinks if necessary. (By the term 'flooding' we mean here processing the cells of the fine DEM in the ordered they would fill with water if the entire coarse cell was to be gradually filled with water starting from the lowest point on the cell's boundary and assuming the cell was surrounded by a continuous rising body of water such that disconnected basins within the cell could start filling from separate edges.) A path is a pair of cells connected by a particular sequence of intermediary cells each one of which directly neighbours (including diagonally) the next cell and the previous cell in the sequence. Paths start from the edges of the section (or next to points marked as sea in a land-sea mask) and continue till they meet another edge (or point neighbouring the sea). When a path is finished it is tested to see if its length exceeds a threshold; this rejects short paths that only cross a single corner of the cell and therefore aren't representative of a flow 'across' the cell. If it returns back to the edge from which it started then the greatest perpendicular separation of the path at any point from its starting edge must also exceed a threshold; this rejects paths that flow back to the same edge unless they represent a meander of a significant size. If the path passes these tests then it is accepted as the lowest valid path through the cell. As any false sinks will have been filled by the algorithm while searching for the path, the last point on the path will be the highest (or joint highest) point on the path. The height of this point is then taken to be the new effective hydrological height of the corresponding coarse cell. Various aspects of the algorithm are illustrated in Fig. 3; these are best understood in conjunction with two aforementioned appendices.

The parameters MINIMUMPATHTHRESHOLD and MINIMUMSEPARATIONFROMINITIALEDGETHRESHOLD (whose use is described in Appendix A) are both set to $0.5 \times$ SCALEFACTOR where:

$$\text{SCALEFACTOR} = \text{Number of latitude points in the coarse grid}/\text{Number of latitude points in the fine grid.} \qquad (5)$$

Originally MINIMUMPATHTHRESHOLD was set to $1.0 \times$ SCALEFACTOR to mirror the equivalent parameter in Tarasov and Peltier's method; however it was noted that this resulted in narrow channels running near parallel across the border between two cells being 'blocked' (both cells having much higher effective hydrological heights than the rest of the channel and thus causing errors in the river directions generated from the upscaled orography created). The current value prevents these blockages and ensures at least one of the two cells the channel runs through has the same hydrological height as the rest of the channel. The algorithm given here can be used on both hydrologically conditioned orographies such as HYDRO1k (USGS, 2017) (as used by Tarasov and Peltier (2006)) and normal (unconditioned) orographies (with false sinks).

For this paper we upscale the unconditioned 30-second orography SRTM30 PLUS (Becker et al., 2009) to a 10-minute grid using the effective hydrological height orography upscaling algorithm described above. The orography upscaling process (which need only be run once) takes approximately 25 minutes to run for the entire globe (from which the section for North

America is then extracted) on a single core of a 2015 MacBook Pro laptop. This extracted section then forms another component of the set of height corrections once it has been combined with any existing corrections in this region.

Where existing corrections from the first two techniques were present in North America the lower value out of the existing correction and the upscaled effective hydrological height was used. Once complete, the corrections were converted to relative corrections by subtracting the original unmodified 10-minute orography. While in many places the application of upscaled effective hydrological heights improve North American river paths, in some places the application of this technique introduced new errors. These errors were corrected by a second round of corrections applied by hand. (It was the necessity to verify changes in the river paths after applying effective hydrological heights and make a second round of additional corrections by hand that required significant additional effort which, as noted above, in turn drove our decision to limit the application of the upscaling of effective hydrological heights to North America.)

When these corrections are applied to an orography for a time other than the present day any relative corrections that are beneath ice sheets are temporarily suppressed until the region becomes ice free once more; thus the original unmodified height is always used for ice sheets. The corrected orography for a time in the past, $t$, to which the river carving algorithm is applied in the next section will at a given cell be:

$$h_{\text{corrected orography}} = \begin{cases} h_{\text{working orography}} + \Delta h_{\text{DEM correction}}, & \text{if } h_{\text{glacier}} = 0 \\ h_{\text{working orography}}, & \text{otherwise} \end{cases} \tag{6}$$

where $\Delta h_{\text{DEM correction}}$ is the fixed relative DEM correction for the given cell from the set of relative DEM corrections whose development has been discussed extensively in this section; $h_{\text{corrected orography}}$ is the height of the corrected orography at the given cell for time $t$; $h_{\text{working orography}}$ is the height of the intermediary working orography as defined in the previous section; and $h_{\text{glacier}}$ is again vertical thickness of the ice-sheet in the cell (which is set to zero if the cell doesn't contain an ice-sheet) for time $t$.

## 2.4 False sink removal

In the third step, the problem of false sinks is solved by using an algorithm that carves rivers out of sinks from the sink's deepest point (Metz et al., 2011). (Barnes et al. (2014) gives a good general overview of priority queue based sink filling algorithms including the particularly clear presentation of the river carving algorithm from Metz et al. (2011) that was followed in the writing of the code for this paper). The algorithm of Metz et al. (2011) imitates water draining from an area through a narrow valley that is not resolved in the DEM because the resolution is insufficient. This gives very similar results to sink filling except that it gives better directions for rivers within the sink itself; although this is most likely unimportant for the final result it can aid the visual comparison of river paths. This algorithm produces river directions directly and neither modifies the input orography itself nor needs to produce a modified copy of it (although in the code written for this paper it can if required create such a copy for purely diagnostic purposes). Both this algorithm and the orography upscaling algorithm are based around the abstract data type called the priority queue. In the context of these algorithms a priority queue is a queue where the cells are kept ordered by ascending height. (More generally a priority queue is any queue kept ordered using a given comparison operator.)

In this algorithm the queue is initially filled with land cells that neighbour the ocean. At each step the cell on the head of the queue (with the lowest height) is removed and processed. Its direct neighbours are assigned river directions pointing to the cell and then added to the queue themselves unless they have already had directions assigned to them previously (in which case they are ignored as they were processed previously; the river direction assigned to them is unchanged and they are not added

to the queue again). By following this procedure when the lowest point on the lip of a sink is reached the algorithm will follow the river down to the bottom of the sink marking a river path that carves out of the sink (i.e. flows uphill) from the bottom to the lip of the sink. Within the sink cells not directly neighbouring the exit path will drain towards the bottom of the sink where they will join the exit path. The possibility exists to mark some points as potential true sinks (i.e. real endorheic basins); if these are in a sink then they are treated as the outflow point for that sink; otherwise processing continues normally. However this option

is not used as it has been decided to remove all true sinks completely when generating dynamic river directions. This closes the water balance using the assumption that precipitation in an endorheic basin would eventually end up in a neighbouring non-endorheic basin either through atmospheric recirculation or via a slow seepage of ground water. This assumption is made in the absence of a full model of dynamic lakes; it may be removed if these are treated by further work. In this absence of dynamic lakes it is necessary to make this assumption (or a similar assumption that the water flowing to true sinks can be redistributed

directly into the ocean) as water conservation is critical for multi-millennial transient paleoclimate simulations. The effects of omitting true sinks are effectively the effects of not modelling dynamic lakes; these effects are discussed in Sect. 6.1. It is still useful to include true sinks when generating data for validation against known modern day river direction information (which also includes true sinks).

## 2.5   Upscaling procedure

The fourth step upscales the 10-minute river directions that have been generated to a $0.5°$ grid using a variant of the COTAT+ upscaling algorithm (Paz et al., 2006). This algorithm itself contains three major steps. The grid cells of the finer grid, referred to as pixels are grouped into sections corresponding to the cells of the coarse grid (these sections are then themselves referred to as cells). The first step is to identify the outlet pixel of each coarse grid cell. This is the pixel with the highest cumulative outflow which meets at least one of two criteria. The first criterion is that the path leading to the pixel through the cell (along

the line of greatest overall cumulative flow) satisfies a minimum path length threshold. The second criterion is the pixel drains the largest number of pixels within the cell in question. These are introduced so that the river direction of the cell is determined by the main river flowing through the cell excluding any rivers that just skirt through a corner of the cell unless they have a tributary which drains a large fraction of the cell itself. The second step is to decide a flow direction for each cell. For each cell the flow path is traced downstream from the chosen outlet pixel until its total cumulative flow has increased by a set amount

or it exits from the direct neighbours of the central cell. The river direction points towards the cell where the downstream tracing finishes. This increases the use of diagonal river directions compared to simply choosing the cell that the outlet pixel directly flows into. The third step is to remove the rare situation of crossing river directions by redirecting the flow direction of the cell whose outlet pixel has the lower total cumulative flow into the same cell as the cell whose outlet pixel has the higher total cumulative flow flows into. However this third step is not used in the variant of the algorithm used here. Although rivers

crossing is clearly unphysical if it did occur it would not negatively affect the quality of the upscaling in terms of the mapping of catchments to river mouths. Instead, two scans are run to identify rarely occurring loops in the upscaled flow directions and re-process the cell where they are occurring to remove them; favouring preserving the river with the highest total cumulative flow.

## 2.6 Flow parameters

The flow parameters are determined using a $0.5°$ orography created by upscaling the 10-minute orography for time $t$ by simply averaging the orography values of the fine grid cells within each coarse grid cell. False sinks are removed from this upscaled orography using a conventional pit filling priority flood algorithm (Soille and Gratin, 1994; Wang and Liu, 2006) (without applying a slope across sinks being filled; so filled sinks are perfectly flat) before it is used to derive the flow parameters. The flow parameters are generated using the same procedure as used for the regular HD model with a few modifications. It was observed in preliminary tests that sink filling occurs across a range of landscapes from rugged to flat and thus generating reservoir retention coefficients for cells within filled sinks as if they were always in an extremely flat region could produce overall an unrealistically slow rate of discharge along rivers. Thus when generating retention coefficients if $0 \geq s$ (meaning either that the cell and its downstream neighbour are both filled sink cells or that the river is flowing uphill when considered on the $0.5°$ scale or the region is actually completely flat to within the accuracy of the DEM) then the value of $s$ is replaced with $s = 1.31 \times 10^{-5}$, a slope value that was observed to produce a typical flow rate. This mostly applies to the generation of river flow retention coefficients however may also be relevant for the generation of overland retention coefficients in a few cells. River reservoir retention coefficients use the $\Delta x$ and $s$ values from the orography for time $t$ upscaled to the $0.5°$ grid. Overland flow retention coefficients use present day inner-slope values from the current JSBACH model where those are non-zero along with the $\Delta x$ values for time $t$ ($\Delta x$ is different for cells with river directions parallel/perpendicular to the grid and those with river directions at an oblique angle to the grid and thus varies with time); otherwise they are generated by a similar technique to the river flow retention coefficients using data from time $t$ only. Base flow retention coefficients use a similar approach; using existing data for the present day where available to account for spatial variability otherwise reverting to the original formulation of Hagemann and Dümenil (1998a). This approach to overland flow and base flow retention coefficients is chosen for simplicity; accurate representation of temporal changes in these parameters is not considered to be important provided plausible values are used throughout the transient simulation.

The initial reservoirs for starting a transient paleoclimate simulation are set by adapting the present day initial reservoirs from the existing HD model. Using a set of present day river directions and flow parameters generated by the method presented in this paper, points that are ocean due to land-sea mask differences (possible as present day land-sea masks can vary), lakes, negative flows (possible due to $P - E$ on glaciers) and wetlands are all removed and replaced with the value(s) of the highest (non-removed) direct neighbour or if that is not possible then the global average(s) (for that/each reservoir type). This setup was run for a year for the present day. It is observed that the model reaches equilibrium or very close to equilibrium after half a year's running when no lakes or wetlands are included. The restart file from this run provides starting values for transient paleoclimate simulations using dynamical hydrological discharge after performing the same set of operations as above on it

again (though obviously there are no lakes or wetlands to remove) using the river directions and flow parameters generated from the starting orography and land-sea mask of the transient climate simulation. The initial reservoirs for periodic restarts of a transient paleoclimate simulation (that occur after stopping to recalculate river directions and flow parameters and any other slow processes necessary) are taken from the restart file produced at the end of the previous run segment. If changes in the land-sea mask have created new land then all of the reservoirs in the new land cells are initialised to zero while if changes in the land-sea mask have flooded land then the contents of all of the reservoirs in the flooded land cells are released into the ocean.

## 2.7 Code and performance

Both the sink filling and river carving algorithms and the orography upscaling algorithm are written in (object-oriented) C++ and share a single code base. The river catchments on the 10-minute scale used in figures are also generated simultaneously to river carving by the same code. This is effectively an application of the algorithm of Beucher and Meyer (1992); Beucher and Beucher (2011). The COTAT+ variant used is written in object-oriented Fortran 2003. Other ancillary tasks are performed in Fortran 90 or Python. Both the sink filling/river carving/orography upscaling algorithms and the COTAT+ river direction upscaling algorithm are designed to be easily extendable to other grids (such as the triangular grid of the ICON-ESM (Zängl et al., 2015)). The total run-time of the code required to generate river directions and flow parameters for a given time-slice on a modern desktop PC with a (multi-core) 3.5 GHz Intel processor is about 1 minute. It is clear from these results that the performance of this code presents no significant issues and it will clearly not impede the performance of the coupled climate model simulations in which it is intended to be embedded. Given the short run-time of the code parallelisation was deemed unnecessary.

## 3 Evaluation for the present day

River directions were evaluated using the total cumulative flow and river catchments. An evaluation of the areas of catchments of major rivers derived from the river directions generated from a 10-minute orography using the method presented here shows that in most cases they match those of the manually corrected river directions currently used in JSBACH to within 5%. Evaluation by eye confirms that the catchment shapes are also very similar. All significant disagreements were identified as being due to minor deficiencies in the manually corrected JSBACH river directions by cross checking against the HydroBASINS catchments. HydroBASINS are a part of the HydroSHEDS dataset (Lehner and Grill, 2013). Adjustments were made to discount discrepancies due to uncorrected true sinks in the river directions derived from the 10-minute orography (as noted above some true sink related errors were ignored in the creation of the corrected orography as all true sinks will be removed for actual paleoclimate simulations). Figure 4 shows zoomed sections comparing the catchments of three major rivers chosen as examples for the manually corrected $0.5°$ present day HD model river directions and for those derived from the 10-minute river directions generated from a corrected 10-minute present day orography. While good agreement is generally observed in these three examples a number of differences are clear round the edges of the catchments. Each difference comprising more than one

or two cells has been checked against various sources of hydrological information; in every case the difference is either due to a minor error in the manually corrected JSBACH river directions or the difference lies in an area of desert with no discernible rivers.

The upscaling algorithm upscales catchment areas to within an accuracy of 1–2% or less in almost all cases for the present day river directions. Evaluation by eye shows that catchment shapes are also extremely similar before and after upscaling. An example of the successful upscaling of the catchment of the Mississippi is given in Fig. 5. As can be observed only a few isolated single cell differences occur. Two cases occur where some additional water enters one catchment and is lost from another catchment. However, if this comparison is repeated for river directions generated without true sinks, then these problems disappear. The only significant problem is the upper reaches of the Mekong's catchment being incorrectly directed into the Yangtze, while some water from the Salween is diverted into the Mekong (thus overall the total area of the Mekong's catchment is roughly correct but the Salween's catchment total area is too small and the Yangtze's too great and the actual location of all three rivers' catchments is partially incorrect). This is illustrated in Fig. 5. This problem is due to the COTAT+ algorithm being unable to cope with the three rivers flowing very close to each other in Yunnan Province, China. This could be fixed by allowing non local flows and using an algorithm like the FLOW algorithm (Yamazaki et al., 2009) but this would require considerable modification of the existing JSBACH HD model. Another possibility would be to run both COTAT+ and an algorithm that generates non-local flows and use the latter to identify and remove disconnects in the former by slightly displacing river paths where necessary.

Figure 6 shows a validation of the automatically generated and upscaled river directions against the manually corrected river directions. Although many differences are observed most of these don't affect which outlet drains which area. Differences that result in a significant change in outlet position for a significant area (more than a couple of cells) have been checked against various sources of hydrological information (primarily HydroBASINS); in all cases they are either due to minor errors in the manually corrected JSBACH river directions directions or the differences lie in areas of desert with no discernible rivers (with the exception of differences connected to the Mekong for which the automatically generated and upscaled river directions are erroneous due to a deficiency of the upscaling procedure as previously discussed).

The flow parameters derived for the present day using the method presented here (including the removal of inland sink points) were compared to the those currently in JSBACH by running the model for one year in a standalone setup with rainfall data as a forcing and comparing the total daily discharge into the ocean (including inland sinks in the case of the current model). The results (not shown) show a very close match; the small discrepancies observed are expected as the current JSBACH model includes inland sinks, lakes and wetlands, all excluded in the dynamic HD model presented here.

The present day river directions and flow parameters generated using the method presented in this paper have been applied in a preindustrial-control simulation using the current Coarse Resolution (CR) version of MPI-ESM. The simulation was started from a steady-state simulation obtained after a long (more than 6000 year) spin-up with the manually corrected present day HD model river directions and flow parameters. The results (not shown) indicate only small local changes, especially in surface salinity close to river mouths. The only exception to this was that the total water flux into the Indo-Pacific was increased and the total water flux into the Atlantic was reduced when using dynamic river directions. The reason for this is that the flow into

inland sinks in Asia that was spread evenly around the world's river mouths when using manually corrected HD river directions was now added to rivers flowing into the Indo-Pacific (as inland sinks had been removed). The large-scale circulation remained largely unchanged.

## 4    Application to an LGM simulation

Figure 7 shows a comparison of the $0.5°$ river directions derived by the dynamic HD method presented here using the present day ICE-6G_C orography and the reconstructed LGM ICE-6G_C orography (Argus et al., 2014; Peltier et al., 2015). The main differences from the present day that are observed in North America at the LGM are an expansion of the catchment of the Mississippi to drain a significant area of the ice sheet surface into the Gulf of Mexico and an expansion of the Yukon to drain part of the north-western ice sheet surface into the Pacific Basin. In Eurasia the flow of a large number of rivers in western Siberia and Scandinavia is blocked by the Fennoscandian ice sheet at the LGM. This forces these rivers to flow either west or east along the ice sheet edge (and thus merge to form two very large rivers). To the west this continues until the flow pathway reaches the North Atlantic Ocean at the western end of the ice sheet; to the east the flow pathway eventually makes a short detour south before reaching the Artic Ocean just beyond the eastern end of the ice sheet. Elsewhere on the globe at the LGM rivers simply extend from their present day mouths to the new extended LGM shoreline.

To validate our approach we compared river directions generated with our method for the LGM to river directions generated directly from a fabricated LGM orography on a 30-second grid created by adding the difference between the reconstructed LGM ICE-6G_C orography and the present day ICE-6G_C orography to the present day SRTM30 PLUS orography. Here, we used the ICE-6G_C orographies on a 10-minute grid; we converted the difference between them to a 30-second grid to match that of the SRTM30 PLUS orography by assigning each 30-second cell the value of the 10-minute cell it would lie within were the 10-minute grid overlaid on the 30-second grid. (This resulted in a blocky structure to the resultant fabricated orography). We then applied the river carving algorithm as described in Sect. 2.4 directly to the fabricated 30-second orography and compared the catchments of the rivers produced to those produced by applying our method to the reconstructed LGM ICE-6G_C orography.

Examination of the results indicates no significant differences in the catchments produced in regions near the ice-sheets with the exception of two changes in the region of Alaska. The first of these is simply the combination of two adjacent river mouths to a single river mouth in the catchments generated from the fabricated 30-second orography. The second is the loss of some of its catchment by the Yukon near $65°$ N $130°$ W in the catchments generated from the 30-second fabricated orography. Investigation shows this is because fine detail of narrow valleys not present in the present day ICE-6G_C orography or the reconstructed LGM ICE-6G_C orography is 'printed' from the SRTM30 PLUS orography onto the surface of the ice-sheet by the fabrication process used; this fine detail allows a river to flow into a catchment to the north following a river pathway in the underlying orography rather than west into the Yukon as it does in the river directions generated using our dynamic HD method. Given the considerable thickness of the ice-sheet at this point it is likely this would not occur physically but the detail of the underlying orography would be smoothed over by the ice-sheet.

To test the effect of dynamically modelling river directions at the LGM against the approach typically used in climate model simulations of this time-slice of simply extending the present day rivers to the new shoreline, two simulations were performed using the boundary conditions from the MPI-ESM LGM simulation of Klockmann et al. (2016). Both simulations integrated the same model as for the present day experiments discussed above using the restart files from Klockmann et al. (2016) but the river direction file differed between the two simulations. One used dynamic river directions generated as described in this paper using the ICE-6G_C orography reconstruction; the other simply extended the present day river directions (including inland sink points) used in JSBACH as standard to the new coastlines. This is consistent with the PMIP3 approach (Braconnot et al., 2011, 2012) for coupled LGM simulations.

Analysis of the two runs is based on climatologies of the last 500 years. Figure 8 shows the difference in freshwater flux into the ocean between the two simulations (including both river outflow and $P - E$ over the ocean surface) on the ocean grid. Figure 9 shows the total freshwater flux into the Indo-Pacific and Atlantic basins as an integrated total from the North Pole to each specific latitude (the implied southward ocean freshwater transport). In both basins a number of localised dipoles are observed; these represent minor differences in the position of the mouth of major rivers and will have very little effect on global circulation patterns. The overall freshwater influx into the Atlantic is reduced and the overall freshwater flux into the Indo-Pacific increased when using dynamic river directions; this change is likely at least partially due to the removal of inland sink points. A significant increase in the catchment of the Mississippi (and thus its outflow) occurs with dynamic river directions while to the north the St Lawrence ceases to exist (although a significant amount of water continues to drain off the ice-sheet in this area) thus there is an overall movement of freshwater southwards. As expected the Fennoscandian ice-sheet causes a significant lateral movement of water to its ends when using dynamic river directions. In the Pacific the main change observed is the merging of the Yangtze and Yellow rivers at their mouths when using dynamic river directions; this produces a large peak in the river outflow but this peak is offset by two troughs on either side. With dynamic river directions the outflow from the Yukon is significantly increased and water is diverted from the North American Artic coast to the Northern Pacific coast of North America. In the Indian and southern Pacific basins little overall change is observed though there are several large local dipoles.

The changes in the freshwater input from rivers have a marked effect on the North Atlantic/Arctic climate system. The changes in continental runoff due to using dynamic river directions lead to a substantial increase in the surface salinity not only in the Newfoundland area, but also in the Labrador Sea. This is shown in Fig. 10. Consequences are enhanced convection in the Labrador Sea, enhanced heat release to the atmosphere, reduced winter sea ice cover and a warming of the atmospheric temperature. An increase in the sea surface temperature (SST) of almost $1°$ C is observed in the sub-polar north west Atlantic. In the Norwegian Sea and the Irminger Sea salinity is reduced when using dynamic river directions. The enhanced stability then reduces convection and the upward mixing of heat in the ocean to the surface. The consequences are a reduction in the SST by about $1°$ C and enhanced sea ice cover.

These changes in freshwater flux forcing have also consequences for the ocean circulation. In the North West Atlantic, the subtropical gyre expands northward in the western half of the basin and the sub-polar gyre becomes weaker and contracts when

using dynamic river directions. However these changes have only a negligible effect on the Atlantic meridional overturning circulation.

## 5 Application to a selected sequences of times during deglaciation

As a demonstration of the modelling of the dynamic evolution of river pathways in North America by the technique presented here, we show in Fig. 11 the major rivers and the most important catchments as generated by the technique for a sequence of four times selected from the last deglaciation . The ice sheet height and isostatic adjustments are taken from ICE-6G_C while the land-sea mask is generated using the technique given in Meccia and Mikolajewicz (2018).

## 6 Discussion and conclusions

### 6.1 Limitations

Limitations of the dynamic river routing technique presented in this paper include the lack of dynamic lakes and wetlands (thus requiring the removal of all true sinks), the lack of sill erosion, and the poor performance of the upscaling algorithm where several major rivers flow in parallel in close proximity.

It was decided to omit dynamic lakes from our method for two reasons. Firstly, the inclusion of dynamic lakes would need direct alteration of the existing HD model code (as opposed to simply altering the input file which contains the river directions and flow parameters); as a component of a ESM this would likely require a considerable quantity of technical work. Secondly, our method cannot distinguish between false sinks and true sinks; the corrections we apply should considerably reduce the number of false sinks in the orographies we use but will not eliminate them. Further processing could solve this second issue but would require the development of further tools. The direct effect of the omission of lakes on the freshwater flux into the ocean will be an inability to model outburst floods that may have played an important role in sudden climate change events such as the Younger Dryas (Rooth, 1982; Broecker et al., 1989). The reduced outflow that occurs when lakes sometimes become closed basins (e.g. Lewis et al., 2001), either because a previous outlet has been blocked and the enclosed basin formed is yet to completely fill with water or because they have a negative water balance because of evaporation, will also be missed. Indirectly the omission of lakes may affect the climate through missing lake-atmosphere interactions (Hostetler et al., 2000; Krinner et al., 2004) and precludes both the inclusion of the mass of the water in the lakes as a feedback to the viscoelastic earth model and the modelling of lacustrine calving of ice-sheets where they are in direct contact with an adjacent lake.

Linked to the lack of lakes (and their associated outburst floods) our model lacks sill erosion; such changes in sill height could affect the preferred outlet of enclosed basins. For example considerable erosion of the southern outlet of Lake Agassiz occurred (e.g. Fisher, 2005); the difference between the current sill height and previous higher sill heights may have had a deciding effect on which outlet overflowed in earlier phases of the lake's development. Wickert (2016) argues that in the case of Lake Agassiz as spillways were usually incised after an outlet outflowed it is likely isostatic adjustments and physical blocking by the ice sheets were the primary drivers of watershed rearrangement during the deglaciation. However, given the complex history of

Lake Agassiz it is possible some outlets may have overflowed at several separate times during the deglaciation thus partly invalidating this argument.

Another important limitation is the lack of verification for time-slices other than the present day; the orography corrections made are largely aimed at producing the correct present day river directions from a present day orography but it is possible that some features of the orography may be unimportant for present day hydrology but critical for hydrology at other points in the last glacial cycle. This is partly addressed by the use of an orography upscaling technique for North America.

Inaccuracies in the orographies of times in the past may also occur due to the model used for calculating isostatic corrections. There are a variety of approaches to viscoelastic earth modelling with differing assumptions (Whitehouse, 2009; Spada et al., 2011); errors from simplified schemes in particular could affect river routing. When using this method as part of a ESM coupled to an ice-sheet model, errors in the simulated size and thickness of the ice-sheet will be passed onto the viscoelastic earth model and thus may drive changes in the river routing that deviate considerably from those observed historically. The degree to which inaccuracies in the underlying orographies of times in the past affects river routings (either because of 'latent' inaccuracies in the present day orography or inaccuracies in the isostatic corrections used to transform the present day orography to orographies of times in the past) is not clear and presents itself as a possible topic for further study.

A further limitation is the sudden step change in the application of orography corrections from ice-free ground (where orography corrections are applied) to the surface of the ice-sheet (where orography corrections are suppressed until the area becomes ice-free again). This may be unrealistic in the case of a thin ice-sheet which will likely continue to follow the contours of the land below it including any narrow valleys which are not resolved in the 10-minute DEM and thus require orography corrections. It is unclear if this would ever have a deciding influence on the routing of any important river pathways. In Sect. 4 the addition of fine detail of the underlying orography affected the Yukon catchment at the LGM; however, it is not clear how physically plausible this fine detail being observed on the surface of the ice-sheet was in this case given the thickness of the ice-sheet were it occurred.

## 6.2 Conclusions

The method presented here provides an effective procedure for the generation of dynamic river directions and flow parameters for paleoclimate simulations. Individually both of the key elements of the method, the application of relative height corrections to a fine orography and the upscaling of a fine set of river directions to a coarse one, have been shown to function to within the required level of accuracy. A special set of relative orography corrections has been used for North America derived using an orography upscaling technique based on the one used successfully by Tarasov and Peltier (2006). Overall when the method presented here is applied to the present day it reproduces the results of a fixed present day hydrological discharge model to a high level of accuracy and all significant discrepancies have been shown either to be in very dry regions or due to minor errors in the fixed river directions (in further comparison to a more detailed set of present day river catchments) or to have negligible effect on the point freshwater is discharged into the ocean. The only exception to this is a problem occurring with the upscaling of the Yangtze, Mekong and Salween rivers in Yunnan province, China. The method is computationally fast enough to be run frequently as part of a wider model reconfiguration process during coupled paleoclimate simulations.

When used in a non-transitory simulation of the present day climate it has been shown that the differences in the ocean system that occur using dynamic river directions and flow parameters compared to the existing fixed river directions and flow parameters are not substantial and limited to localised salinity changes. It has been shown that using dynamic river directions and flow parameters has a significant effect on the water flux to the ocean when applied to the LGM, increasing outflow from the Mississippi and redirecting water from the Mackenzie into the Yukon on the ice-sheet itself along with a major lateral movement of freshwater to the ends of the Fennoscandian ice-sheet. Coupled simulations for the LGM indicate that these changes in the freshwater flux entering the ocean have a significant effect on the global ocean circulation through changes to the North Atlantic/Artic climate system and these effects are also transferred to the atmosphere.

In summary, we have shown that modelling changes in hydrological discharge is important for modelling ocean circulation at the LGM and have presented a method by which changes in hydrological discharge can be modelled for transient coupled climate model simulations of the last glacial cycle.

## 7  Code availability

A version of the code is available under the 3-Clause BSD License on Zenodo at https://doi.org/10.5281/zenodo.1326547. This omits elements of the flow parameter generation code discussed in section 2.6 that are part of the existing HD model's parameter generation code and must be excluded for licensing reasons. A complete version of the code is stored within the JSBACH 3 model repository in the Apache version control system (SVN) of the Max Planck Institute for Meteorology (https://svn.zmaw.de/svn/cosmos/branches/mpiesm-landveg/contrib/dynamic_hd_code/) at revision 9313 under the Max Planck Institute for Meteorology Software License Version 2. For access to this complete version of the code (including the omitted elements) apply to the lead author.

## 8  Data availability

The final set of relative height corrections (as discussed in section 2.3) is available as a NetCDF file under the Creative Commons Attribution 4.0 License on Zenodo at https://doi.org/10.5281/zenodo.1326394.

## Appendix A:  Outline of the orography upscaling algorithm

The algorithm's structure is based on that of the priority flood algorithm (Soille and Gratin, 1994; Wang and Liu, 2006) but however requires substantial modification from this original basis to carry the extra information required for orography upscaling and to accommodate the necessity of sometimes going back along sections of previously rejected paths from the opposite direction in order to explore all possible paths. Central to this algorithm is the priority queue abstract data type, as described in Sect. 2.4. An outline of the algorithm is given here; a more formal description using pseudo-code is given in Appendix B. (In addition, a flow diagram illustrating the steps of the algorithm is given in the supplementary material.) The algorithm comprises the following steps:

1. Split the fine gridded orography into sections, each of which corresponds to one cell of the coarse orography. This is illustrated in Fig. 3(a). This step corresponds to lines 4 and 11 of algorithm 1 in the pseudo-code description.

2. Loop over the sections. For each section of the fine orography calculate an effective height and then replace the height of the coarse orography cell that section corresponds to with this effective height. This step corresponds to lines 10-19 of algorithm 1 in the pseudo-code description. The effective hydrological height of each section is calculated thus:

    (a) *This step prepares the initial content of the priority queue we will later iterate over.* Push each cell from along the section's edges onto a priority queue ordered by cell height. Also push all cells neighbouring cells marked as sea in a fine scale land-sea mask onto the queue. This is illustrated in Fig. 3(b). This first set of cells added to the queue are henceforth referred to as initial cells. (Sea cells themselves are not added to the queue here or elsewhere in this algorithm. Note when using a fine scale land-sea mask the land-sea boundaries are not limited to running along section boundaries hence the necessity of adding their neighbours explicitly.) In the following description we refer to the path leading to a particular cell as that cell's path; paths can be of any length greater than zero - sometimes these paths comprise only the cell itself. For each cell store the cell's height, its position, a unique identifier of the starting edge of the cell's path (which will be the edge the cell is on for initial cells), a path length value set to 1 for initial cells (or $\sqrt{2}$ if the cell is a diagonal neighbour of a sea cell), the farthest separation of the cell's path from its initial edge (which is set to zero for initial cells), a unique identifier of the cell's pseudo-catchment (a unique identifier of the starting point of the cell's path - which for initial cells will simply be a unique identifier of the cell itself) and the initial height of the cell's path (the height of the starting point of the cell's path - which naturally for initial cells will be the height of the cell itself except if it is the neighbour of a sea point in which case it will be sea level). This step corresponds to algorithm 2 in the pseudo-code description.

    (b) *This step sets up storage arrays for variables that need to be stored as a spatial field. This completes the initialisation.* Setup a boolean array flagging cells already processed with the same dimensions as the section. Mark as processed in this array cells neighbouring cells marked as sea; mark all other cells as unprocessed. Setup two arrays with the same dimensions as the section to contain the unique identifiers of the cells' pseudo-catchments and the initial heights of the cells' paths. This step corresponds to lines 6–8 and 14–15 of algorithm 1 in the pseudo-code description.

    (c) *This step starts a loop over the contents of the priority queue; unless we break from the loop each iteration spans from this step to the end of step (e). In this step itself we fetch the next cell to be processed from the queue and update one of its properties.* Pop the lowest height cell off the queue. Calculate the separation of this cell from its path's initial edge and update the farthest separation of the cell's path from its initial edge with this new value if it is greater than the current value. Mark the cell as processed in the boolean array flagging cells already processed. This step corresponds to lines 2–4 of algorithm 3 in the pseudo-code description.

    (d) *This step checks if the current cell is the end of a valid path (which will by design be the lowest valid path) through the fine orography section. If it is we use its height as the effective hydrological height of the corresponding coarse*

*cell and move on to processing the next section of the fine orography.* If the cell is an edge cell or neighbours a cell marked as sea in the land-sea mask then check if the cell satisfies the parameter MINIMUMPATHTHRESHOLD. If it has returned to the same edge that its path started from then check if the parameter MINIMUMSEPARATIONFRO-MINITIALEDGETHRESHOLD is satisfied. If the check passes (or both the checks pass if the second check was also made) then take the height of this cell as the effective height for the section and finish processing this section and move to the next iteration of the loop opened in step 2. Examples of paths failing the check(s) are given in Fig. 3(c) while examples of paths passing the check(s) are given in Fig. 3(d) and 3(e). This step corresponds to lines 5–12 of algorithm 3 in the pseudo-code description.

(e) *This step iterates over the neighbours of the current cell, skipping those that have already been processed unless they require reprocessing. Each neighbour that has not been skipped has its properties updated as appropriate and is added to the priority queue; along with being marked as processed. Also values that would be necessary for potential future reprocessing are written to the appropriate spatial storage arrays at the neighbour's position. Reprocessing is required when the algorithm has been working progressively on two separate pseudo-catchments that have now met at a certain point. In order to explore all the possible paths it is necessary for one of these catchments to be reprocessed and added to the other catchment starting from the meeting point. The reprocessed catchment should be the one that started from a higher initial point to ensure that all paths run from a lower point to a higher one (even if they pass even higher points still en route). Here two criteria are used to correctly enact such a reprocessing when it is required.* Loop over all the neighbours of the cell (as illustrated in Fig. 3(c)). Hereinafter we referred to the cell whose neighbours are being considered as the centre cell. For each neighbouring cell:

    i. If the neighbouring cell is not marked as already processed in the boolean array flagging processed cells then continue to step ii without making any checks. If the neighbouring cell is marked as already processed then check if:

      – The neighbour's initial path height (as read from the array of the initial path heights of processed cells) is greater than that of the centre cell. In the case that the path heights are equal then use a tie-breaking criterion based on the unique identifiers of the centre cell's and neighbouring cell's pseudo-catchments to decide if to skip the neighbour or not; this prevents infinite loops. The unique identity of the neighbouring cell's pseudo-catchment is read from the array of the unique identifiers of cells' pseudo-catchments.

      – The neighbour's path started from a different point to the centre cell's (also based on the unique identifiers of the centre cell's and neighbouring cell's pseudo-catchments).

    If both these criteria are met then continue to step ii; otherwise skip processing this neighbour and move on to the next iteration of the loop.

    ii. Mark the neighbour as processed in the array of processed cells.

iii. Write (overwriting previous values where necessary) the unique identifier of the centre cell's pseudo-catchment and the initial height of the centre cell's path to the respective arrays of those variables at the neighbour's position.

iv. Push the neighbouring cell onto the queue using the values of the centre cell apart from path length and cell height which are both replaced with new values. For path length a new value is calculated by adding the distance from the centre cell to the neighbouring cell (either $1$ or $\sqrt{2}$ if it is a diagonal neighbour) to the centre cell's existing path length and for cell height whichever is higher out of the centre cell's height and the neighbour's height is used (thus for cell height the new value may be the same as the old value).

Once all the neighbours have been processed return to step (c). This step corresponds to lines 13–26 of algorithm 3 in the pseudo-code description.

## Appendix B: Orography upscaling algorithm pseudo-code

The main body of the algorithm is given as pseudo-code in Algorithm 1 while two important sub-algorithms used by the main algorithm are given in Algorithms 2 and 3. In these algorithms we use $\leftarrow$ to denote the assignment operator and $=$ to denote a test for equality. Variables written in italicised camel case are containers; specifically either arrays, annotated cell objects or a priority queue. Italicised lower case variables (with or without a subscript) are simply numbers (or coordinates in the case of *pos*) while words in full sized capitals are function names and words in small capitals are either externally supplied parameters or constants/identifiers. Words in lower case bold represent flow control structures (if statements, while loops, for all loops, return statements and sub-algorithm calls) or logical operators. Brackets represent the initialisation of an object/structure using a group of variables with given values unless they are: positioned directly after an array variable, in which case they represent indexing of that array using the position indicator enclosed within the brackets; are used in an if statement, in which case they indicate the order of operations; or they are positioned after a function, in which case they enclose arguments to the function. The variables $C_{\mathrm{C}}$, $C_{\mathrm{F}}$, *pos* and $N$ store coordinates within a DEM grid that locate a cell within that DEM; note these are different from annotated cell objects which allow the storage of further information about a cell in addition to its position. For $C_{\mathrm{F}}$, *pos* and $N$ these are positions within a fine scale orography; for $C_{\mathrm{C}}$ these are positions within the coarse DEM to be produced by this algorithm. These positions can be used to index arrays.

At input *FineDEM* is an array of orography on a fine scale and *FineLandSeaMask* an array of the land-sea mask on the same scale using possible states LAND and SEA (in practice usually represented by a boolean array) while *CoarseDEMDimensions* are the required dimensions of the coarse DEM to be produced. Also required are values for the parameters MINIMUMPATHTHRESHOLD and MINIMUMSEPARATIONFROMINITIALEDGETHRESHOLD and a value for the constant SEALEVEL representing the sea-level datum to be used (normally zero). RIVERMOUTH, TOPEDGEID, BOTTOMEDGEID, LEFTEDGEID and RIGHTEDGEID must be unique identifiers. NODATA is a simple null value used to fill array elements for which a value is yet to be calculated. At output *CoarseDEM* is an orography of effective hydrological heights on the given coarse scale.

*Author contributions.* The manuscript was prepared by Thomas Riddick with substantial contributions from Uwe Mikolajewicz in Sect. 3 and 4. Thomas Riddick designed the orography upscaling algorithm and wrote the code specific to this project. Part of the code used for the generation of retention coefficients was adapted from that used in the HD model developed by Stefan Hagemann. Coupled modern day, deglaciation and LGM climate simulations were performed by Uwe Mikolajewicz. All authors contributed to the development of ideas and scientific direction.

*Competing interests.* The authors declare that they have no conflict of interest.

*Acknowledgements.* We acknowledge funding by the German Federal Ministry of Education and Research (BMBF) funded project PalMod. We thank Florian Ziemen for providing the code used for plotting ocean variables and for reviewing the manuscript prior to submission.

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

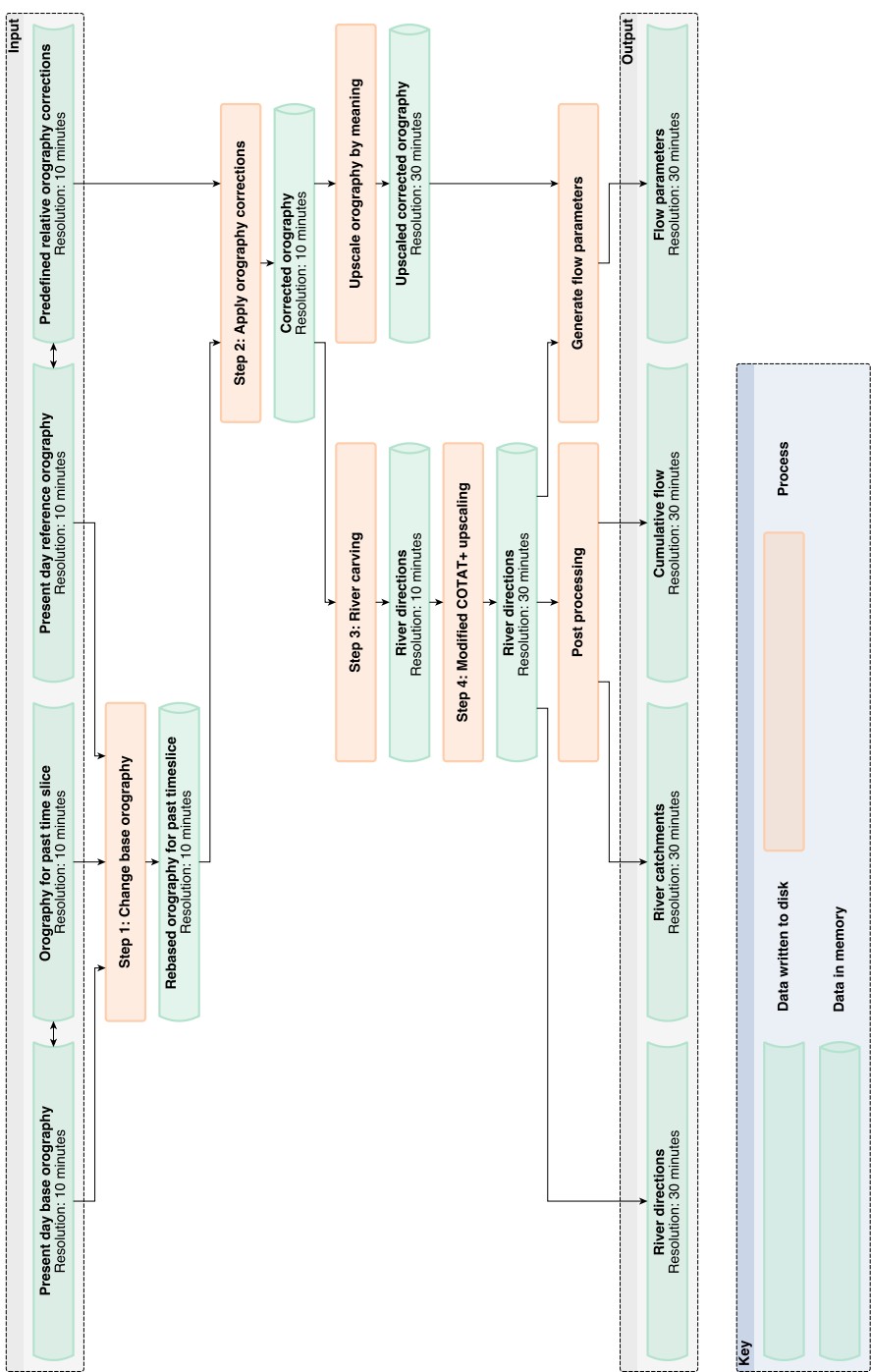

**Figure 1.** Flow diagram illustrating the steps of the method presented here for generating river directions and flow parameters for dynamic hydrological discharge modelling. (Here "upscale orography by meaning" means simply taking the mean value of the nine 10-minute DEM cells contained within the area covered by each 30-minute DEM cell as the value of that 30-minute DEM cell.)

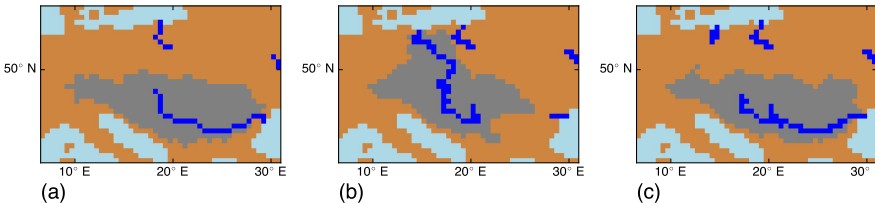

(a)          (b)          (c)

**Figure 2.** Comparison of the Danube catchment showing the catchment (grey area) and rivers with a total cumulative inflow greater than or equal to 75 cells (blue cells) derived from (a) manually corrected $0.5°$ HD model river directions as a reference, (b) automatically generated river directions for a 10-minute grid and (c) automatically generated river directions for a 10-minute grid once height corrections have been applied to a few selected cells in the orography.

---

**Algorithm 1** Orography upscaling algorithm main

Compare to steps 1 and 2 of the description given in Appendix A.

---

**Require:** *FineDEM*, *FineLandSeaMask*, *CoarseDEMDimensions*

1: Let *AnnotatedCell* be an object that can store the position $pos$ and height value $height$ of one cell in *FineDEMSection* along with a number of variables describing the path to that cell, namely: the height of the cell at the start of the path $initialheight$, the identity of the edge at the start of the path $edgeid$, the length of the path $len$, the greatest perpendicular distance the path travels from the starting edge $greatestperpdist$ and a catchment number unique to all paths starting from the same point $catchid$

2: Let *Open* be a priority queue of *AnnotatedCell* objects with total order ordered by $height$ (the height of the cell)

3: Let *CoarseDEM* have dimensions equal to *CoarseDEMDimensions*

4: Let *FineDEMSection* have dimensions equal to the dimensions of *FineDEM* divided by the dimensions of *CoarseDEM*

5: Let *FineLandSeaMaskSection* have the same dimensions as *FineDEMSection*

6: Let *Closed* have the same dimensions as *FineDEMSection*

7: Let *PathInitialHeights* have the same dimensions as *FineDEMSection*

8: Let *CatchmentNumbers* have the same dimensions as *FineDEMSection*

9: Let *CoarseDEM* be initialised to NODATA

10: **for all** coarse cells $C_\mathrm{C}$ in *CoarseDEM* **do**

11:     *FineDEMSection* $\leftarrow$ section of *FineDEM* corresponding to $C_\mathrm{C}$

12:     *FineLandSeaMaskSection* $\leftarrow$ section of *FineLandSeaMask* corresponding to $C_\mathrm{C}$

13:     Reset *Open* to be an empty queue

14:     *Closed* $\leftarrow$ **false** where *FineLandSeaMaskSection* is LAND

15:     *Closed* $\leftarrow$ **true** where *FineLandSeaMaskSection* is SEA

16:     $count \leftarrow 0$

17:     **call** Setup Queue (see Algorithm 2)

18:     **call** Process Queue (see Algorithm 3)

19: **end for**

20: **return** *CoarseDEM*

---

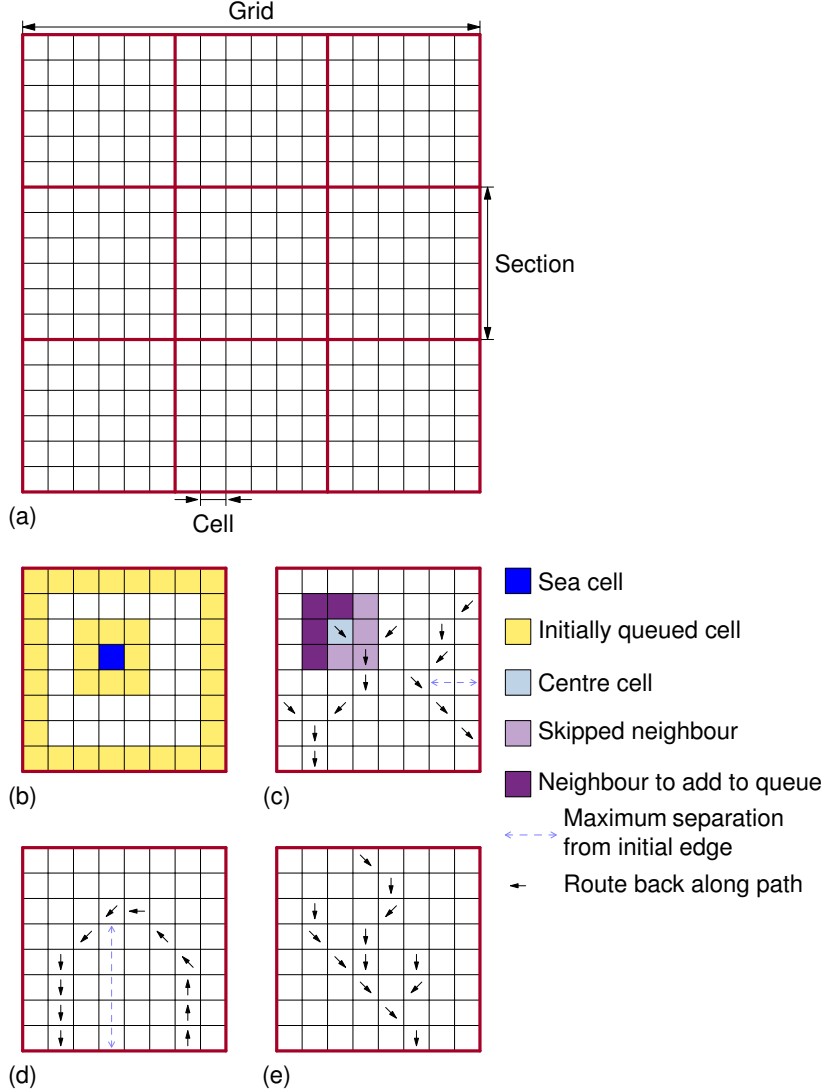

**Figure 3.** Diagrams illustrating various aspects of the orography upscaling algorithm. In (a) the division of a DEM grid into sections is shown; the upscaling algorithm processes each section separately to produce an effective hydrological height for each section. (b) shows the initial cells (see main text of Appendix A) added to the queue at the start of the algorithm including the neighbours of a sea point. (c) shows three paths: two complete but rejected because they don't meet the selection criteria (see main text) for a valid lowest path, one incomplete. The short path in the bottom left corner is complete but its length is too short for it to qualify as the lowest valid path through the cell; the longer path on the right is also complete but it returns to the same edge it started at without having met the required maximum separation from the initial edge threshold. The path in the middle, which branches from the short path in the bottom left, is incomplete and a cell at its end is undergoing processing. Half this cell's neighbours have been added to the queue; the other half have been skipped because they have already been processed. In (d) we show a valid lowest path through the cell that returns to its initial edge but meets both of the selection criteria while in (e) we show a valid lowest path through the cell that spans two different edges and has several incomplete paths branching off it.

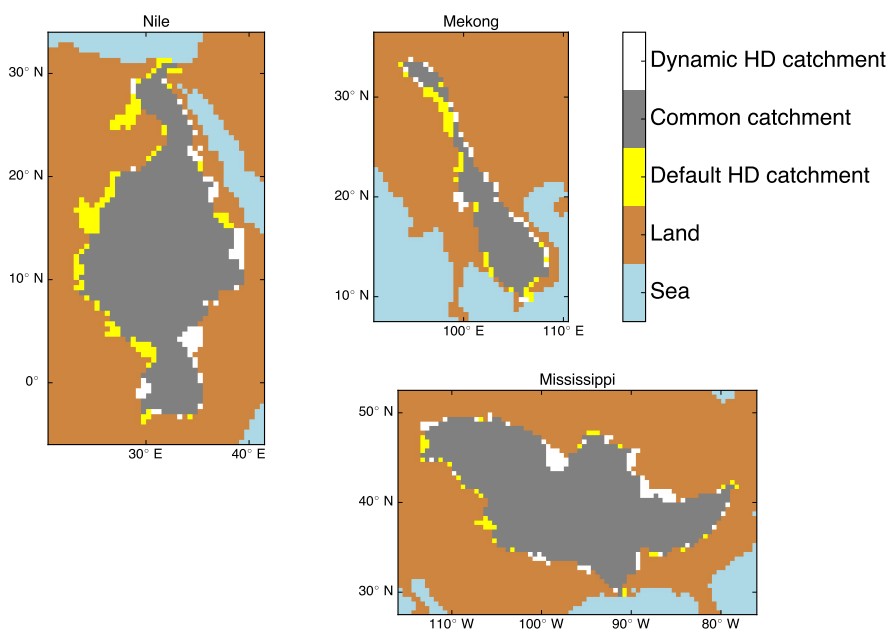

**Figure 4.** Comparison of the manually corrected $0.5$ degree catchments for the present day ("Default HD") to those of the 10-minute directions generated from a corrected 10-minute present day orography ("Dynamic HD") for the Nile, Mekong and Mississippi.

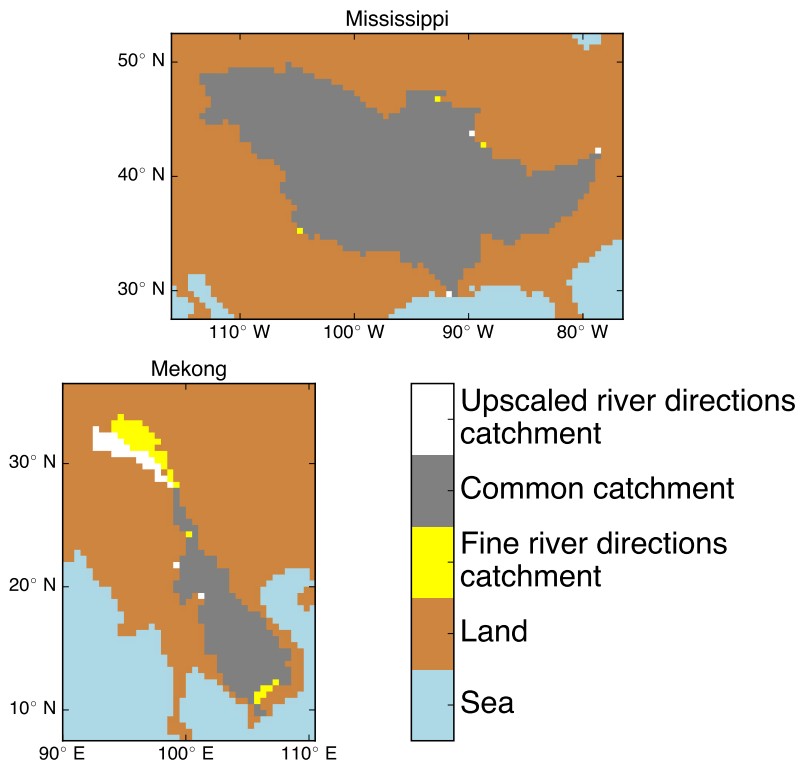

**Figure 5.** Comparison of the upscaled catchments of the Mississippi and Mekong on a 0.5° grid to the original 10-minute version.

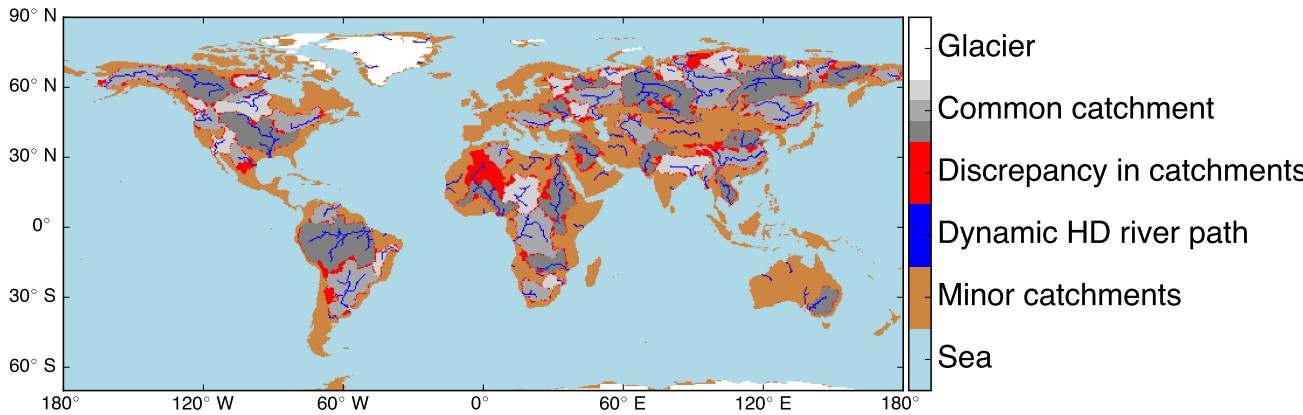

**Figure 6.** Comparison of the most significant present day river catchments derived using the method presented here ("Dynamic HD") to the manually corrected HD model river directions. Discrepancies are shown in red; areas of catchments that are common between both models are marked in grey. The three different shades of grey are used to pick out individual river catchments; no significance is attached to the shade chosen for each river. Dynamic HD river paths (defined as cells with a cumulative flow of 100 cells or more) are marked to aid orientation.

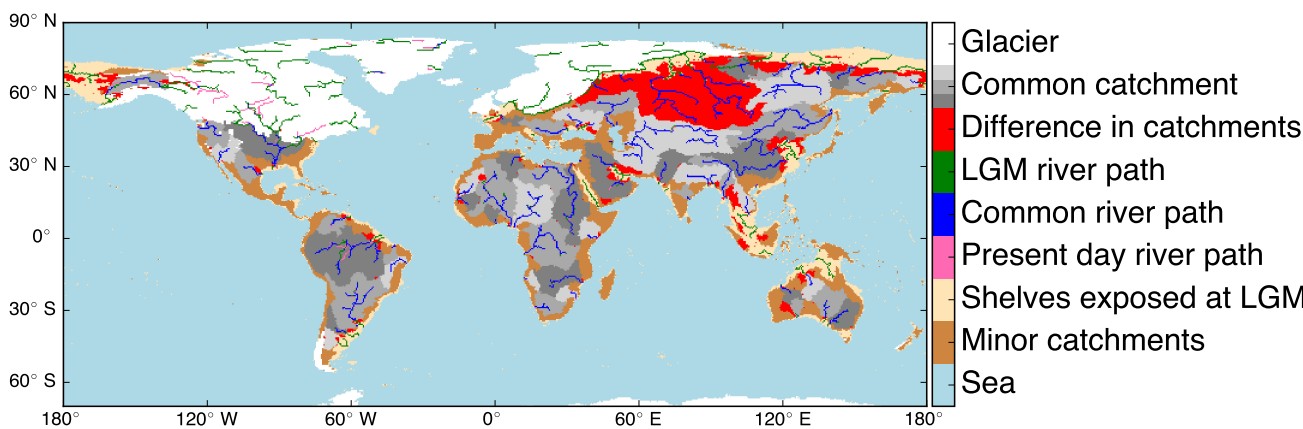

**Figure 7.** Comparison of the most significant rivers at the LGM and present day generated by the method described in this paper using the ICE-6G_C reconstructed orography for the LGM. Rivers are shown where the cumulative flow to a given grid cell (on the HD grid) is greater than or equal to 100 cells. The various colours show various rivers that existed only at the LGM (green), only at the present day (pink) or at both (blue). The catchments of major rivers are marked. Difference between the catchments are shown in red; areas of catchments that are common between both time-slices are marked in grey. The three different shades of grey are used to pick out individual river catchments; no significance is attached to the shade chosen for each river. Continental shelves which were exposed as dry land at the LGM by the significantly lower sea-level are also marked.

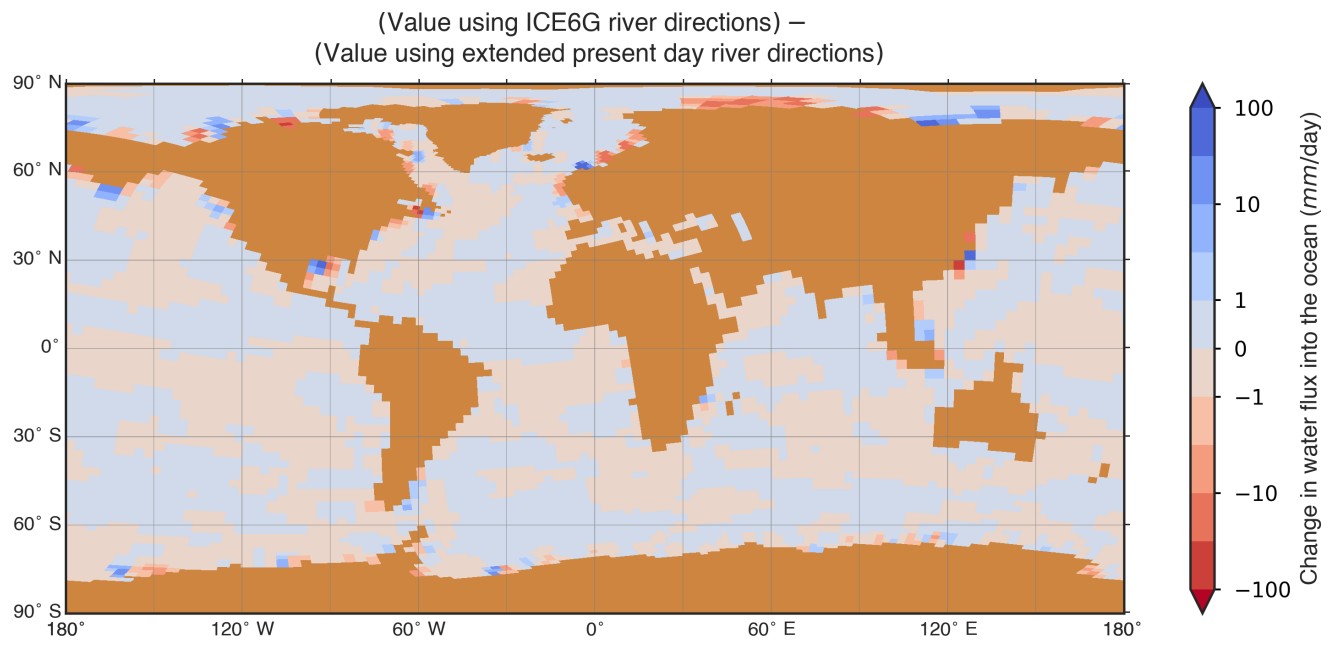

**Figure 8.** Changes in the freshwater flux into the ocean between simulations run in the MPI-ESM model of the LGM using extended present day river directions and using dynamic river directions. The changes are defined such that an increase in the version using dynamic river directions is positive. A symmetrical logarithmic colour scale is used: above 1 the colour scale is logarithmic; between 1 and −1 the colour scale is linear; below −1 the colour scales according to the negation of the logarithm of the change's magnitude.

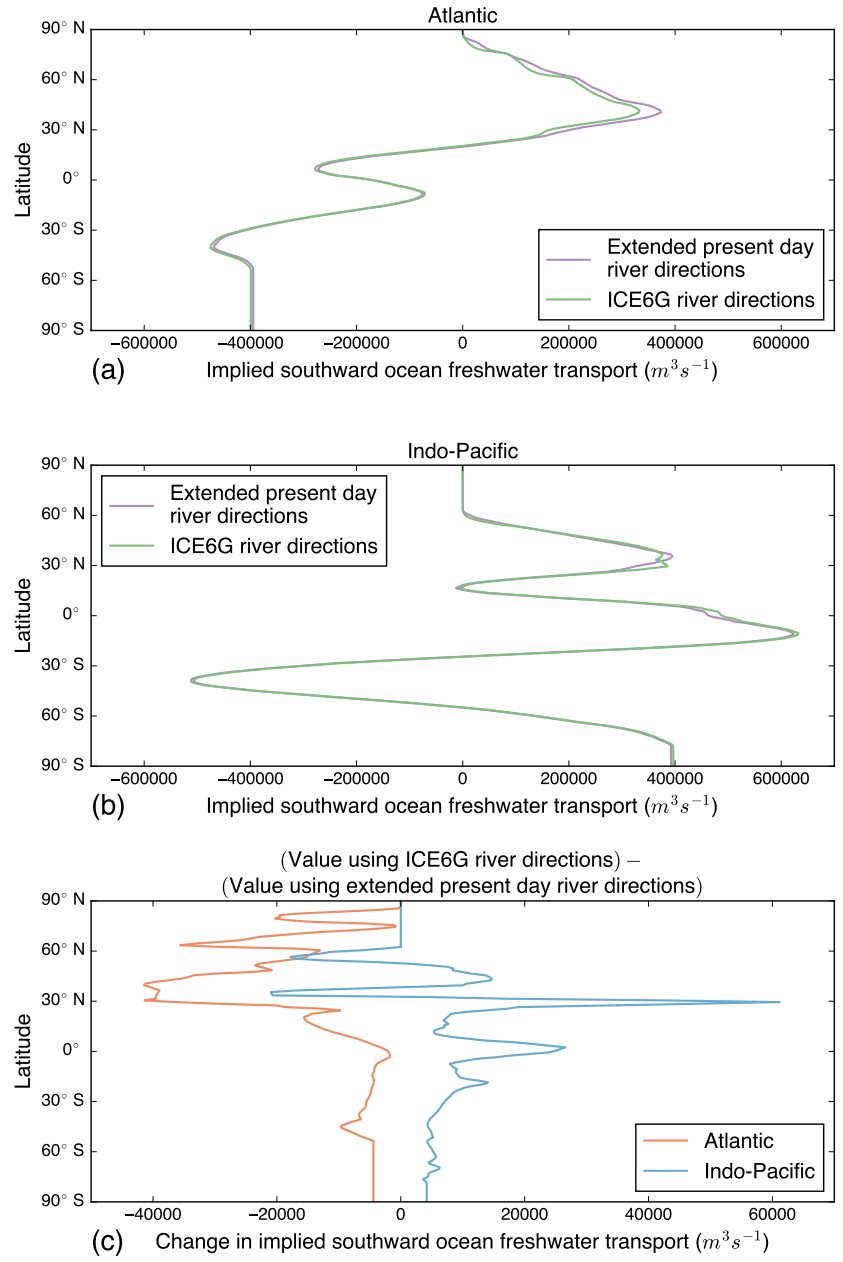

**Figure 9.** Comparison of the implied southward ocean freshwater transport between simulations run in the MPI-ESM model of the LGM using extended present day river directions and using dynamic river directions for (a) the Atlantic Ocean and (b) the Indo-Pacific Ocean. Plot (c) gives the difference between the two simulations for both basins. The freshwater transport is defined such that a net addition of freshwater to the ocean (via precipitation and river discharge) is positive and a net removal of freshwater (via evaporation) is negative.

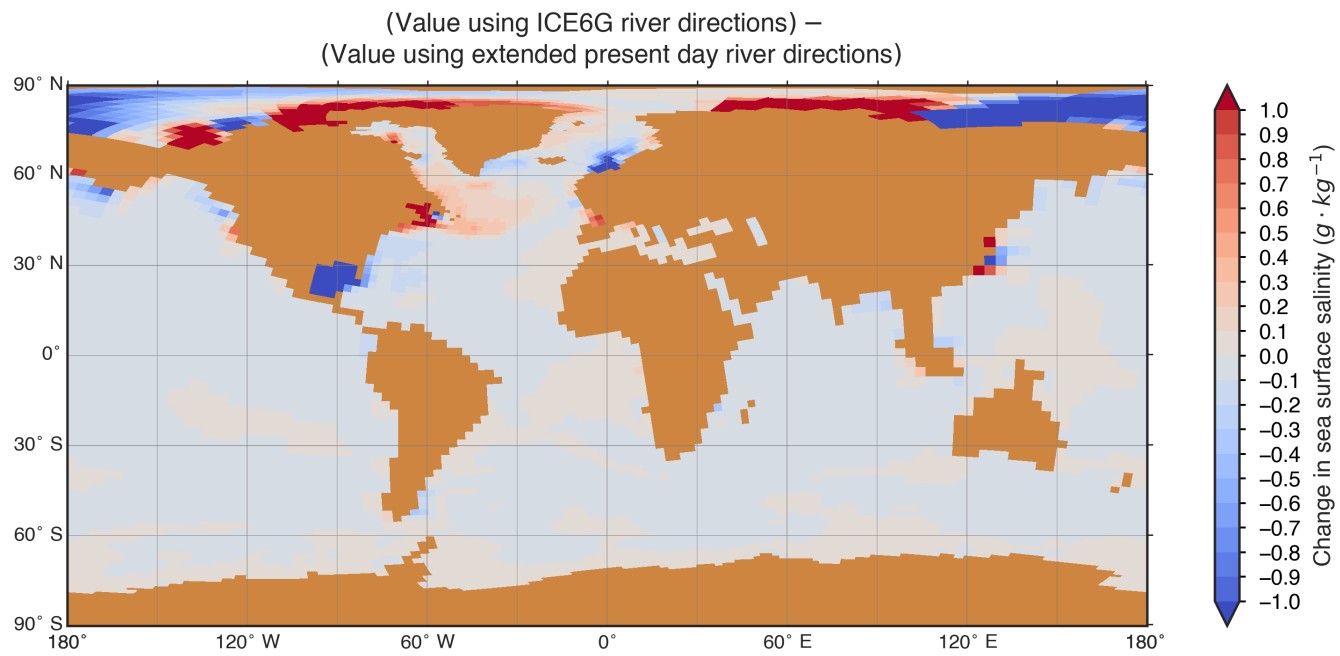

**Figure 10.** Changes in the surface ocean salinity between simulations run in the MPI-ESM model of the LGM using extended present day river directions and using dynamic river directions. The changes are defined such that an increase in the version using dynamic river directions is positive.

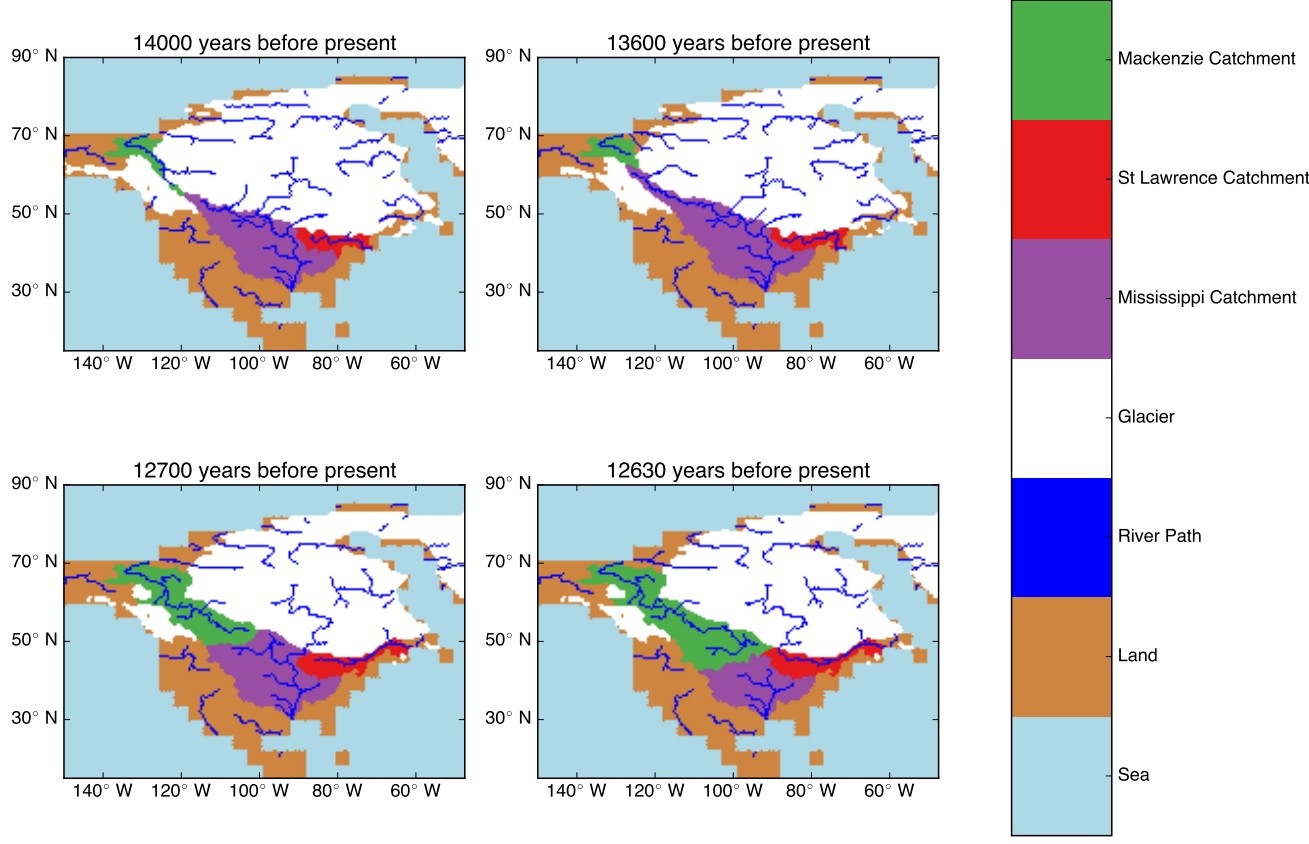

**Figure 11.** Comparison of rivers generated using the method presented here for four times during the last deglaciation using the ICE-6G_C orography reconstruction. Rivers are shown where the cumulative flow to a given grid cell (on the $0.5°$ grid) is greater than or equal to 75 cells. The catchments for the Mississippi, St Lawrence and Mackenzie rivers are marked. Note the diversion of the St Lawrence to a different mouth point for the two older times.

**Algorithm 2** Orography upscaling algorithm setup queue

Compare to step 2(a) of the description given in Appendix A.

---

**Require:** *FineDEMSection*, *FineLandSeaMaskSection*, *PathInitialHeights*, *Open*, *CatchmentNumbers*, *Closed*, *AnnotatedCell*, *count*

1: **for all** fine cells $C_\text{F}$ in *FineDEMSection* neighbouring cells where *FineLandSeaMaskSection* is SEA **do**

2:     **if** *FineLandSeaMaskSection*($C_\text{F}$) is SEA **then**

3:       **skip to next iteration of loop**

4:     **end if**

5:     *AnnotatedCell* $\leftarrow$ ($pos \leftarrow C_\text{F}$, $height \leftarrow$ *FineDEMSection*($C_\text{F}$), $initialheight \leftarrow$ SEALEVEL, $edgeid \leftarrow$ RIVERMOUTH, $len \leftarrow 1$ **if** $C_\text{F}$ is a non-diagonal neighbour of a cell where *FineLandSeaMaskSection* is SEA **else** $len \leftarrow \sqrt{2}$, $greatestperpdist \leftarrow 0$, $catchid \leftarrow count$)

6:     Push *AnnotatedCell* onto *Open*

7:     *CatchmentNumbers*($C_\text{F}$) $\leftarrow$ count

8:     $count \leftarrow count + 1$

9:     *PathInitialHeights*($C_\text{F}$) $\leftarrow$ SEALEVEL

10:     *Closed*($C_\text{F}$) $\leftarrow$ **true**

11: **end for**

12: **for all** fine cells $C_\text{F}$ on the edges of *FineDEMSection* **do**

13:     **if** *FineLandSeaMaskSection*($C_\text{F}$) is SEA **then**

14:       **skip to next iteration of loop**

15:     **end if**

16:     *AnnotatedCell* $\leftarrow$ ($pos \leftarrow C_\text{F}$, $height \leftarrow$ *FineDEMSection*($C_\text{F}$), $initialheight \leftarrow$ *FineDEMSection*($C_\text{F}$), $edgeid \leftarrow$ TOP/BOTTOM/LEFT/RIGHTEDGEID(as appropriate), $len \leftarrow 1$, $greatestperpdist \leftarrow 0$, $catchid \leftarrow count$)

17:     Push *AnnotatedCell* onto *Open*

18:     *CatchmentNumbers*($C_\text{F}$) $\leftarrow$ count

19:     $count \leftarrow count + 1$

20:     *PathInitialHeights*($C_\text{F}$) $\leftarrow$ *FineDEMSection*($C_\text{F}$)

21: **end for**

22: **return** *Open*, *Closed*, *PathInitialHeights*, *CatchmentNumbers*

---

**Algorithm 3** Orography upscaling algorithm process queue

Compare to steps 2(c)-(e) of the description given in Appendix A.

---

**Require:** *FineDEMSection*, *FineLandSeaMaskSection*, *PathInitialHeights*, *Open*, *CatchmentNumbers*, *Closed*, *AnnotatedCell*, $C_C$, *CoarseDEM*

1: **while** *Open* is not empty **do**
2:     $C_F$, $height_{CEN}$, $initialheight_{CEN}$, $edgeid_{CEN}$, $len_{CEN}$, $greatestperpdist_{CEN}$, $catchid_{CEN} \leftarrow$ POP(*Open*)
3:     $greatestperpdist_{CEN} \leftarrow$ MAX($greatestperpdist_{CEN}$, current perpendicular separation from initial edge $edgeid_{CEN}$)
4:     $Closed(C_F) \leftarrow$ **true**
5:     **if** $C_F$ is an edge cell **or** $C_F$ neighbours one or more cells which are SEA in *FineLandSeaMaskSection* **then**
6:         **if** $len_{CEN} >$ MINIMUMPATHTHRESHOLD **then**
7:             **if** (**not** (identity of edge at $C_F$) $= edgeid_{CEN}$) **or**
            ($greatestperpdist_{CEN} >$ MINIMUMSEPARATIONFROMINITIALEDGETHRESHOLD **and** **not** $edgeid_{CEN} =$ RIVERMOUTH)
            **then**
8:                 $CoarseDEM(C_C) \leftarrow height_{CEN}$
9:                 **exit loop**
10:             **end if**
11:         **end if**
12:     **end if**
13:     **for all** neighbouring cells $N$ of $C_F$ **do**
14:         **if** $Closed(N)$ is **true then**
15:             **if** *PathInitialHeights*$(N) < initialheight_{CEN}$ **or**
            *CatchmentNumbers*$(N) = catchid_{CEN}$ **or**
            (*PathInitialHeights*$(N) = initialheight_{CEN}$ **and** *CatchmentNumbers*$(N) > catchid_{CEN}$) **or**
            *FineLandSeaMaskSection*$(N)$ is SEA)) **then**
16:                 **skip to next iteration of loop**
17:             **end if**
18:         **end if**
19:         *FineDEMSection(N)* $\leftarrow$ MAX(*FineDEMSection(N)*, $height_{CEN}$)
20:         $Closed(N) \leftarrow$ **true**
21:         *CatchmentNumbers*$(N) \leftarrow catchid_{CEN}$
22:         *PathInitialHeights*$(N) \leftarrow initialheight_{CEN}$
23:         neighbour path length $len_N \leftarrow (len_{CEN} + 1)$ **if** $N$ is a non-diagonal neighbour of $C_F$ **else** $len_N \leftarrow (len_{CEN} + \sqrt{2})$
24:         *AnnotatedCell* $\leftarrow$ ($pos \leftarrow N$, $height \leftarrow$ *FineDEMSection(N)*, $initialheight \leftarrow initialheight_{CEN}$, $edgeid \leftarrow edgeid_{CEN}$,
        $len \leftarrow len_N$, $greatestperpdist \leftarrow greatestperpdist_{CEN}$, $catchid \leftarrow catchid_{CEN}$)
25:         Push *AnnotatedCell* onto *Open*
26:     **end for**
27: **end while**
28: **return** *CoarseDEM*