# Peer review of "Dynamic hydrological discharge modelling for coupled climate model simulations of the last glacial cycle: The MPI-DynamicHD Model version 3.0"

_Geoscientific Model Development, 2018_

## Short Comment (SC1) · 4 Apr 2018

ear authors,

in my role as Executive editor of GMD, I would like to bring to your attention our Editorial version 1.1: http://www.geosci-model-dev.net/8/3487/2015/gmd-8-3487-2015.html This highlights some requirements of papers published in GMD, which is also available on the GMD website in the 'Manuscript Types' section: http://www.geoscientific-model-development.net/submission/manuscript_types.html In particular, please note that for your paper, the following requirement has not been met in the Discussions paper:

[Figure]

- "The main paper must give the model name and version number (or other unique identifier) in the title."

Please provide the name and a version number of your method for dynamic hydrological discharge modelling in the title of your revised manuscript. Note, that a name and a version number are important to identify your development (name) and its state (version number).

Best regards,
Astrid Kerkweg
* * *

---

## Author Comment (AC1) · 16 Apr 2018

Thank you for pointing this out to us. We will ensure this is dealt with in the final version of the manuscript.

Best Regards,

Thomas Riddick (on behalf of all of the authors)

---

## Referee Comment (RC1) · L. Tarasov (Referee) · 7 May 2018

The submission "Dynamic hydrological discharge modelling for coupled climate model simulations of the last glacial cycle" presents a routing toolbox/module for coupled paleoclimate modelling. The latter is necessarily of coarse resolution relative to river and catchment scales and the challenge (from this reviewer's own experience) of appropriate upscaling from hydrologically resolving Digital Elevation Models is very non-trivial. The module is clearly validated in Figure 6. It's dynamic downslope approach makes this a useful component for any paleoclimate Earth Systems Model (ESM), especially given the growing interest in running ESMs with fully coupled ice sheets.

The only two significant but easy to address deficiencies I found were the lack of lakes and no specific mention of other dynamic sources of uncertainty that need to be considered for accurate paleo-drainage modelling, specifically glacial isostatic adjustment (GIA) and the need to consider erosional changes in controlling outlet sills.

I don't understand why the algorithm can't handle lakes. Large pro-glacial lakes were significant deglacial feature for Europe and especially North America, of relevance to both the climate (eg evaporative surface) and to the adjacent ice sheet (lacustrine calving margin). There is significant evidence in the glacial geological literature (eg, Fisher, 2005), that at least the southern outlet for Glacial Lake Agassiz experienced significant erosion and therefore lowering of the controlling sill depth over the lifetime of that lake. Higher older sills can force routing into other ocean basins with potentially significant climatic consequences. The paper should at least provide a brief explanation of why lakes aren't computed and the impacts thereof.

Fisher, T.G., 2005. Strandline analysis in the southern basin of glacial Lake Agassiz Minnesota and North and South Dakota, USA. Geological Society of America Bulletin 117 (11/12), 1481–1496.)

GIA and determines geographic and geoidal deflections and therefore errors from common simplified GIA schemes can also potentially significantly affect drainage routing. Again, all that is needed is a short statement to this effect. I note the authors do mentions "inaccuracies in the underlying topography" as a issue, but I think it would be useful to readers to spell out the key dynamical sources of this.

My only other comment (which may be due to my ignorance of conda) relates to all the listed version specific required libraries in the code archive dynamic_hd_env.txt. I suspect this was auto generated and I would strongly urge the authors to reduce this to the bare bones required. Ideally, code should port without special language features, and the only versions issues should be to avoid known bugs in specific versions.

I have some minor to moderate comments below, but once all this is addressed, I would

judge this a worthwhile contribution to the community.

**specific comments (quotes lack the "#" prefix)**

on which an existing present-day hydrological discharge model within the JSBACH land surface module

**I'm confused here. You describe this as "present-day", so presumably just a routing matrix for present-day topography, but then below you indicate this does down-slope routing, in which case this does not need the "present-day" qualifier.**

has also been shown this procedure can be run successfully multiple times as part of a transient coupled climate model simulation.

**shouldn't this be run continuously in async mode? Ice sheet margins can significantly change drainage routes within a century.**

During the last glacial cycle, the courses of rivers in North America, ..

**should provide appropriate references**

however such lakes are switched off entirely in the version of the HD model used for dynamic hydrological discharge modelling by this paper

**why? Can the not be run with the dynamic upscaled topography?**

can be accurately reproduced on a 0.5˚o resolution

**For ice sheet modelling, the meridional grid convergence towards the poles means that it makes much more sense to use eg 0.5 longitude by 0.25 latitude degree resolution.**

We define the effective hydrological height of a cell within a DEM as the height of the highest point in the 'most likely' river

**more succinct description would be "elevation of river sill within the cell"**

Upscaling effective hydrological heights could also potentially be applied beneficially to Eurasia however we decided against doing so because of the significant additional effort required.

**I'm confused here as I understood from the reading that the core upscaling process is automated, in which case the effort is negligible. Farther down I note**

The orography upscaling process (which need only be run once) takes approximately 25 minutes to run for the entire globe

**so why not use the generated global field?**

When these corrections are applied to an orography for a time other than the present day any relative corrections that are beneath ice sheets are temporarily suppressed until the region becomes ice free once more; thus the original unmodified height is always used for ice sheets

**How do you handle the transition? Eg, say 200 m thick ice in the Rocky Mountains will not over-ride the existing topography?**

Another important limitation is the lack of verification for time-slices other than the present day (due to the lack of easily comparable data);

**This is an arguable statement. One could take the LGM topographic deflection from GIA, apply it to the hydro-1k DEM, overwrite ice topography, and then use eg the R hydrological flow solver (or equivalent in other GIS apps) to extract drainage maps and compare to your results.**

algorithmic pseudocode

**I found this hard to follow. eg it's not clear what all the variables such as p, d, h, i represent. As long as all the code is provided, I would suggest simplifying the pseudo code somewhat and use more descriptive names than single letter indices. Try running the pseudo code by some colleagues and see what doesn't make sense to them.**

[Figure]

Code archive and code quality

**I tried to set up the code set, but the required conda environment manager is not an available (apt-get accessible) package for Linux and installing this along with what I take are a lot of specific versions of python libraries.. was more than I was willing to go.**

Figure 1,

**what does upscale by "meaning" mean?**

Figure 8,

**would be much more informative to show change in volume flux eg in deci Sverdrups than change in water velocities.**

**small typos:**

Artic Coast -> Arctic coast

Coast -> coast

northern Pacific -> Northern Pacific

---

## Referee Comment (RC2) · D. Yamazaki (Referee) · 24 May 2018

<General Comment> The manuscript describes a method to calculate river flow directions dynamically. This must be of interest to hydrologist and paleo-climate researchers, given that the preparation of the paleo river network is difficult. The paper can be potentially acceptable, but there are some major issues to be addressed. 1. Provide a literature review on the method for generating paleo river direction map. There is no citation to the previous method in the manuscript. If there is no previous study, please state so. 2. The method section is difficult to understand. The authors stated that the method is applied for every 10-100 years, but the method description

suggested some manual processing is needed. It is not clear how the authors could update the river direction dynamically during the paleo climate simulation. 3. Related to above, only one river direction for the paleo climate was shown, though the method must have been applied multiple times during the simulation. As long as the authors stated that the method is for "dynamic river direction", the time series of gradual river direction change should be shown as a figure.

iijIJSpecific Comments> P2. L26: Negative values of are set to a constant. Please clarify which value was used. I guess, zero slope is also problematic, this the authors actually used "minimum threshold". Please clarify.

P3. L4: The challenge for a paleoclimate simulation is to develop a method for periodically updating the river directions and flow parameters used with sufficient accuracy. Please provide reference to previous papers. How LGM river map was prepared in past studies. Is there any previous method which can treat "dynamic" river map generation?

P3. L20: False sinks also appear at higher resolutions due to various imperfections in the measurement of orography satellite: Recommended citation to the errors in DEM is: Yamazaki D., D. Ikeshima, R. Tawatari, T. Yamaguchi, F. O'Loughlin, J.C. Neal, C.C. Sampson, S. Kanae & P.D. Bates A high accuracy map of global terrain elevations Geophysical Research Letters, vol.44, pp.5844-5853, 2017 doi: 10.1002/2017GL072874

P4:L8 A brief outline is given here From this section, it seems all steps are automated. While in Section 2.3 the authors mentioned the "by-hand" method which must be not automated. Please clarify this discrepancy. The "by-hand" correction was applied only once at the first step? Then, the description in Section 2.1 should be revised to avoid confusion.

P4. L19: any intervening cells It is not clear that what "intervening cell" means.

P4. L24: 2.2 Changing the present day base orography This section is very difficult to understand. What is the difference between the present-day-base-orography amd
present-day-reference-orography? Do they have different characteristics? Which data is used as "present-day-base-orography"? Do the two DEMs have the spatial resolution? Please clarify.

P5. L15: effective hydrological heights It is not clear what is the "effective hydrological height". Please write a brief explanation when it first appears.

P6. L2: below a given threshold What threshold was used? Please clarify.

P6. L23: as the height of the highest point in the most likely river pathway It is not clear why "highest point" was used. Please explain.

P6: L25: lowest highest point This expression is confusing. Please revise.

P7. L1: "flooding each coarse cell" It is not clear what "flooding" means here. Please explain in detail.

P7. L13: If the detailed description here is not needed to explain the main work (regenerating river maps during the climate simulation) proposed in the manuscript, I recommend to move this section to Appendix.

P10. L8: For this paper we upscale the unconditioned 30-second orography SRTM30 PLUS (Becker et al., 2009) to a 10-minute grid Is this used as "present day base orography" in Eq(3)? It is not clear how this data is used.

P13. L19: as all true sinks will be removed for actual paleoclimate simulation Is this a reasonable assumption? Please discuss.

P14. L14: in all cases they are either due to minor errors in the manually corrected JSBACH river directions The error could be "minor" for the purpose of climate simulation, but the same error could be critical for different purpose. The authors have to acknowledge the method is suitable for climate simulations, but the accuracy could be not adequate for different use (such as water resource assessment or flood risk modelling).

GMDD
P14. L31: Application to a LGM simulation P14.L32: Figure 7 I guess, many readers want to check how the algorithm calculates gradual change in river directions following the change in orography. Given that the authors stated in Section 2.1 that "River directions are regenerated approximately every 10-100 years", there must me multiple river maps for glacier period simulations. However, the authors showed only 1 river map for the paleo climate simulation. I think this is not adequate to proof the usefulness of the proposed method. I here suggest to show the time series of river map development, by focusing on some locations where the authors can observe the gradual change of river directions during the paleo simulation. Otherwise, it is difficult to state that the proposed method is "dynamics" river direction mapping.

Figure 7: Please use different color for the "new land" or "new ocean" in Figure 7 to show the change in land sea mask.

---

## Author Comment (AC2) · 4 Jun 2018

We thank Lev Tarasov for his useful comments. We give our initial responses below; we will give further responses and add corrections to the manuscript shortly.

1. Comment: The only two significant but easy to address deficiencies I found were the lack of lakes and no specific mention of other dynamic sources of uncertainty that need to be considered for accurate paleo-drainage modelling, specifically glacial isostatic adjustment (GIA) and the need to consider erosional changes in controlling outlet sills.

I don't understand why the algorithm can't handle lakes. Large pro-glacial lakes were

significant deglacial feature for Europe and especially North America, of relevance to both the climate (eg evaporative surface) and to the adjacent ice sheet (lacustrine calving margin). There is significant evidence in the glacial geological literature (eg, Fisher, 2005), that at least the southern outlet for Glacial Lake Agassiz experienced significant erosion and therefore lowering of the controlling sill depth over the lifetime of that lake. Higher older sills can force routing into other ocean basins with potentially significant climatic consequences. The paper should at least provide a brief explanation of why lakes aren't computed and the impacts thereof.

Response: We acknowledge the importance of lakes in palaeohydrology in both North America and Europe. A dynamic lake model is planned and a prototype of it is under active development. However, this work is likely to be ongoing for some time and given the schedule of the PalMod project (for which this tool is primarily intended) and also the time required to integrate changes to components successfully into a wider paleoclimate Earth System Model the present hydrological discharge model without dynamic lakes will receive a significant amount of usage in scientifically important runs. We will alter the manuscript to include a consideration of the impact of the omission of lakes and an explanation of the reasoning for this omission. We further clarify the reasoning for omitting them below.

There are two barriers to the inclusion of periglacial lakes in our model.

a) Apparent basins in the orography can be due to either false sinks (narrow river valleys), lakes or unfilled/partially unfilled endorheic basins (perhaps best thought of as 'potential lakes'). Neither a pit filling nor a river carving algorithm can distinguish between these. Our method of applying height corrections to the orography to give the correct present-day river directions will considerably reduce the number of blocked narrow river valleys but not eliminate all of them as in some cases the present-day river finds the correct route despite a narrow valley not resolved in the orography and thus no error occurs for the present day and no corrections are applied – however what was not an error for the present day may be an error for a past time slice if the tilt of
the landscape has been changed by isostatic rebound. The technique of upscaling effective hydrological heights is likely even more effective at eliminating such blocked narrow valleys but will still not work in 100% of cases as errors can still occur and if in this case the present-day river finds the correct path regardless the need for corrections is again not apparent. Subsequent to the submission of this paper we have developed a new tool which will ensure that for the present day all false sinks have a 1 cell wide downslope path to the sea while respecting the basins of actual lakes and endorheic catchments and ensuring they have the correct sill height; from this a further set of relative corrections will be made and added to those that already exists for use in the generation of paleohydrologies. However as for the technical reasons to be discuss next we aren't able to include lakes in the present setup this tool is beyond the scope of this paper.

b) The hydrological discharge model or method presented in this paper doesn't require any modifications to the actual code of JSBACH itself but works by generating new input files for JSBACH every time the model is stopped and restarted (every 10 model years). As mentioned with the existing HD model of JSBACH there is already a lake model but this is unsuitable for modelling lakes whose size and depth changes or which cover multiple grid boxes; a lake cell in this existing model is in effect a single reservoir with an infinite holding capacity and a set average residency time for water - this is effective for modelling the buffering effect of modern day lakes on hydrological discharge but not for modelling lakes with dynamically changing sizes. To introduce a new lake model will require the HD model within JSBACH to be modified; like many GCM components JSBACH was not design to cope with changing input data and thus this modification is likely to require a significant technical development within the model. We agree that erosion of lake sills has a potentially important impact on paleo-hydrology and hope to address this in the lake model we are now developing. We will add a comment noting this lack of sill erosion as a source of error in the text and discuss the effects of this.
2. Comment: Fisher, T.G., 2005. Strandline analysis in the southern basin of glacial Lake Agassiz Minnesota and North and South Dakota, USA. Geological Society of America Bulletin 117 (11/12), 1481–1496.) GIA and determines geographic and geoidal deflections and therefore errors from common simplified GIA schemes can also potentially significantly affect drainage routing. Again, all that is needed is a short statement to this effect. I note the authors do mentions "inaccuracies in the underlying topography" as an issue, but I think it would be useful to readers to spell out the key dynamical sources of this.

Response: Hypothetically our method can be used with any GIA scheme. Within PalMod it will be used with the VILMA model. We will add a short statement noting the GIA scheme used is a potentially important source of error and discuss the sources of this; both those in the original topography as a representation of the present day and those due to the imperfect transformation of that topography to represent a past time including accounting for erosion.

3. Comment: My only other comment (which may be due to my ignorance of conda) relates to all the listed version specific required libraries in the code archive dy-namic\_hd\_env.txt. I suspect this was auto generated and I would strongly urge the authors to reduce this to the bare bones required. Ideally, code should port without special language features, and the only versions issues should be to avoid known bugs in specific versions.

Response: Although this list was indeed automatically generated all of these libraries are required directly or indirectly by some element of the either the dynamic hydrological discharge code or the plotting code that goes with it. Extensive use of external libraries is typical for scientific python code. Many are included in the very extensive python standard library but some are part of a number of third party libraries. Python itself has a number of dependencies on other packages: openssl, readline, sqlite, tk, zlib, pip, setuptools and wheel. Scipy (which Numpy is part of; see http://www.numpy.org) is de facto the standard toolkit for scientific computing with Python. netcdf4-python is a
package that allows netCDF4 files to be loaded and written in python. It has a lot of dependencies on other packages; namely: hdf5, libnetcdf, numpy, gfortran, zlib, curl, mkl, krb5, openBLAS. Matplotlib is a very popular plotting package for Python (designed to emulate MATLAB's functionality) and also has a number of dependencies, namely: cycler, freetype, libpng, numpy, pyparsing, pyqt, python-dateutil, pytz, six, qt, sip and openssl. Also used for plotting is the backend cairo, on which depends fontconfig, libxml2, pixman and pycairo. We could create separate environments for the plotting code and reduce this list of packages somewhat however it would still be long enough with the netCDF4 module and its dependencies to require the use of a package manager and with the Conda package manager using the Anaconda repository the setup of these libraries should be done automatically. The Conda package manager and the associated Anaconda repository is used as it is a recommended method of installing Matplotlib according to Matplotlib's own install instructions and the top listed method of installing Scipy on the Scipy website. Its use allows close control over the python environment setup to ensure the correct version of each package is installed.

4. Comment: # I'm confused here. You describe this as "present-day", so presumably just a routing matrix for present-day topography, but then below you indicate this does down-slope routing, in which case this does not need the "present-day" qualifier.

Response: Prior to this work on dynamic hydrology we had a fully static hydrology in the HD model in JSBACH with a set of prescribed river directions for the present day – this is still the HD model used for non-paleo work with JSBACH. The river directions for this were derived manually in several steps. First the orography was pit-filled; then a downslope routing was generated. The river directions thus generated were then themselves corrected by hand to ensure the correct paths for the world's main rivers and further corrections to the river directions were made based on a careful comparison of the catchment of each major river to reference catchments to ensure that each catchment was as far as possible correct.

5. Comment: # shouldn't this be run continuously in async mode? Ice sheet margins
can significantly change drainage routes within a century.

Response: We will most likely run with a 10-year interval. True continuous running would be technically very difficult to implement within the MPI-ESM structure and this method is not fast enough to be run in such a way. It would be possible to run this method in parallel to provide a half yearly update to the river routing with a half year delay/offset however this would also be technically problematic as we would either have to stop and start the main ESM model every half year or allow it to let the river directions change during the course of a run. The former would likely be computationally inefficient and the latter very complicated to implement. Also, even if we implemented it would only be possible to update the ice margins every half year unless we also ran the GIA model VILMA as well which would be a further computational expense. We note the other referee has said our explanation of how the method is applied is unclear; thus, we will include a short explanation of how a paleoclimate ESM models will be/are run in transient mode – i.e. a series of 10-year model runs each using the starting conditions of the previous run and calculating river routing, GIA adjustments and the changing land-sea mask between runs.

6. Comment: "During the last glacial cycle, the courses of rivers in North America, .." # should provide appropriate references

Response: Agreed; we will add one to the final text

7. Comment: "however such lakes are switched off entirely in the version of the HD model used for dynamic hydrological discharge modelling by this paper" # why? Can the not be run with the dynamic upscaled topography?

Response: The lakes provided in the existing HD model of JSBACH are not suitable for a changing topography; they can only be single grid cell in size (multiple lake grid cells can be placed together but they are independent of one another) and their effect is to buffer throughflow. This gives a good representation of non-endorheic lakes for the present day but basin-filling lakes covering multiple grid cells will need a completely
new model. As noted above this is future work and lies beyond the scope of this paper. The current lake models buffering effect was not seen as useful for paleolakes as we would have had to use present day parameters which would not be accurate for paleohydrologies. Given that the exact timing of discharge is not of great importance for the kind of coupled climate model paleo simulations this model is intended for it was thus preferred to drop the existing lake model entirely.

8. Comment: # For ice sheet modelling, the meridional grid convergence towards the poles means that it makes much more sense to use eg 0.5 longitude by 0.25 latitude degree resolution.

Response: We need to use 0.5-degree grid to match that of the existing Hydrological Discharge model code in JSBACH3. On the one hand, implementing a 0.5 by 0.25-degree resolution would require significant technical changes to the JSBACH3 code itself; something this method has so far avoided as experience shows that changes to the land surface model itself are often time consuming to implement even when apparently simple in theory. On the other hand, the generation of a the present-day model orography and river direction on the HD 0.5 degree grid has required a significant amount of time, as many corrections are necessary to derived a realistic and consistent orography from an existing orography dataset (see our response to comment 4 and Hagemann and Dümenil 1998).

9. Comment: "We define the effective hydrological height of a cell within a DEM as the height of the highest point in the 'most likely' river" # more succinct description would be "elevation of river sill within the cell" C3

Response: Agreed, we will change to this wording.

10. Comment: "Upscaling effective hydrological heights could also potentially be applied beneficially to Eurasia however we decided against doing so because of the significant additional effort required." # I'm confused here as I understood from the reading that the core upscaling process is automated, in which case the effort is negligible.
Farther down I note

Response: Yes, it is automated... but see our next comment below

11. Comment: "The orography upscaling process (which need only be run once) takes approximately 25 minutes to run for the entire globe" # so why not use the generated global field?

Response: Despite notably improving the accuracy of the river directions produced the orography upscaling method doesn't remove all errors in present day river directions and some manual corrections are still required and it wasn't possible to make these for the entire globe due to time pressures. It is highly likely we will revisit this topic in future work on lakes and expand the use of this technique to the entire globe; possibly starting from an even higher resolution orography to reduce the number of manual corrections required.

12. Comment: "When these corrections are applied to an orography for a time other than the present day any relative corrections that are beneath ice sheets are temporarily suppressed until the region becomes ice free once more; thus the original unmodified height is always used for ice sheets" # How do you handle the transition? Eg, say 200 m thick ice in the Rocky Mountains will not over-ride the existing topography?

Response: We don't apply any special handling of the transition; we simply add the height of the ice-sheet to the unmodified 10-minute orography when deriving the orography of ice covered regions. We acknowledge this could be a source of error (and will add it to our limitations section in the text) but it is not clear to the authors how to realistically model the growth of ice over the kind of narrow valley not resolved in the unmodified 10-minute orography and at what point an ice sheet growing over it would essentially smooth out this feature. This could potentially be future work although the first step would be to investigate how likely this is to occur and if so what effect it might have. It would likely be only a small impact unless such a thin ice sheet occurred for an extended period at a point critical for the determination of drainage directions.
13. Comment: Another important limitation is the lack of verification for time-slices other than the present day (due to the lack of easily comparable data); # This is an arguable statement. One could take the LGM topographic deflection from GIA, apply it to the hydro-1k DEM, overwrite ice topography, and then use eg the R hydrological flow solver (or equivalent in other GIS apps) to extract drainage maps and compare to your results.

Response: This could be interesting; we will try making this comparison. However, this process is also really a model rather than data; thus, this is a model to model comparison. We hope to make more extensive comparisons to data during future work on lakes; this will obviously be directed mostly at the lakes but necessarily validate also river directions. We will remove "(due to the lack of easily comparable data);" in the final text.

14. Comment: algorithmic pseudocode # I found this hard to follow. eg it's not clear what all the variables such as p, d, h, i represent. As long as all the code is provided, I would suggest simplifying the pseudo code somewhat and use more descriptive names than single letter indices. Try running the pseudo code by some colleagues and see what doesn't make sense to them.

Response: As suggested we will change the single letter indices for more descriptive names and provide comments to better clarify the pseudo-code. The pseudo-code is certainly difficult to understand however we are very hesitant to simplify the pseudo-code as it would no longer provide an essentially complete description of the algorithm and this would then make it very similar to the description provided in the main text. We have actually provided 4 different descriptions of the same algorithm (if we count the code) and it is our intention that each one describes it to a different level of detail and thus with a different level of readability. We would not expect a reader to be able to understand the pseudo-code from a single reading; this is the purpose of the written description of the algorithm in the main text (which is admittedly also complicated but is simplified as far as is possible); instead it functions as a template for anyone trying to
rewrite the code in another language and as a reference to check details against while reading the description in the main text. The code itself (mostly found in fill\_sinks.cpp and sink\_filling\_algorithm.cpp) is very considerably less readable still (despite being documented with comments in the proscribed C++ style), particularly as it shares a code base with two variants of the sink filling/river carving algorithm, has additional options for dealing with true sinks not described in the paper (and not presently used), is designed to be sufficiently abstract to be applicable to any arbitrary grid and as a consequence of these requirements uses the C++ features multiple-inheritance and generic programming (i.e. templates). We hope in the pseudo-code to present a description of the complete algorithm in a 'clean' code-like form without all of this additional technical infrastructure and without any account for issues such as memory management.

15. Comment: # I tried to set up the code set, but the required conda environment manager is not an available (apt-get accessible) package for Linux and installing this along with what I take are a lot of specific versions of python libraries.. was more than I was willing to go.

Response: For reasons unknown to the authors the Anaconda python distribution (of which the open source Conda is the package manager) can't be installed via apt-get. Linux installation instructions are given here: https://conda.io/docs/user-guide/install/linux.html

We will produce another release of the code for the final version of the paper including more extensive instructions on setting up the code and suggestions of how you might install scipy, matplotlib and netCDF4 without using Conda/Anaconda if preferred.

16. Comment: Figure 1, # what does upscale by "meaning" mean?

Response: We mean upscaling by taking the mean value of the height of all the fine DEM cells that are contained within the area covered by a given coarse DEM cell as the average height of that coarse DEM cell. We will clarify this in the caption or on the figure itself if there is room.
17. Comment: Figure 8, # would be much more informative to show change in volume flux eg in deci Sverdrups than change in water velocities.

Response: We will try plotting this and see how it looks. We originally preferred to use a quantity that was per unit area because of the uneven grid cell size (this figure uses the ocean models grid) but there could be an argument for replacing this with deci Sverdrups.

18. Comment: # small typos: ### Artic Coast -> Arctic coast Coast -> coast northern Pacific -> Northern Pacific

Response: Thanks, we will correct these in the final text.

---

## Author Response (AR1)

**Authors' Response**

All page number references are to the latex-diff version of the manuscript.

**General Notes**

We have changed the title of the manuscript to include a model name and version number as requested by GMD executive editor Astrid Kerkweg.

**Response to Comments of Lev Tarasov**

We thank Lev Tarasov for his useful comments.

**1. Comment:** The only two significant but easy to address deficiencies I found were the lack of lakes and no specific mention of other dynamic sources of uncertainty that need to be considered for accurate paleo-drainage modelling, specifically glacial isostatic adjustment (GIA) and the need to consider erosional changes in controlling outlet sills.

I don't understand why the algorithm can't handle lakes. Large pro-glacial lakes were significant deglacial feature for Europe and especially North America, of relevance to both the climate (eg evaporative surface) and to the adjacent ice sheet (lacustrine calv- ing margin). There is significant evidence in the glacial geological literature (eg, Fisher, 2005), that at least the southern outlet for Glacial Lake Agassiz experienced significant erosion and therefore lowering of the controlling sill depth over the lifetime of that lake. Higher older sills can force routing into other ocean basins with potentially significant climatic consequences. The paper should at least provide a brief explanation of why lakes aren't computed and the impacts thereof.

**Response:** We acknowledge the importance of lakes in palaeohydrology in both North America and Europe. A dynamic lake model is planned and a prototype of it is under active development. However, this work is likely to be ongoing for some time and given the schedule of the PalMod project (for which this tool is primarily intended) and also the time required to integrate changes to components successfully into a wider paleoclimate Earth System Model the present hydrological discharge model without dynamic lakes will receive a significant amount of usage in scientifically important runs. We give the reasoning for omitting them below.

There are two barriers to the inclusion of periglacial lakes in our model.

1) Apparent basins in the orography can be due to either false sinks (narrow river valleys), lakes or unfilled/partially unfilled endorheic basins (perhaps best thought of as 'potential lakes'). Neither a pit filling nor a river carving algorithm can distinguish between these. Our method of applying height corrections to the orography to give the correct present-day river directions will considerably reduce the number of blocked narrow river valleys but not eliminate all of them as in some cases the present-day river finds the correct route despite a narrow valley not resolved in the orography and thus no error occurs for the present day and no corrections are applied – however what was not an error for the present day may be an error for a past time slice if the tilt of the landscape has been changed by isostatic rebound. The technique of upscaling effective hydrological heights is likely even more effective at eliminating such blocked narrow valleys but will still not work in 100% of cases as errors can still occur and if in this case the present-day river finds the correct path regardless the need for corrections is again not apparent. Subsequent to the submission of this paper we have developed a new tool which will ensure that for the present day all false sinks have a 1 cell wide downslope path to the sea while respecting the basins of actual lakes and endorheic catchments and ensuring they have the correct sill height; from this a further set of relative corrections will be made and added to those that already exists for use in the generation of paleohydrologies. However as for the technical reasons to be discuss next we aren't able to include lakes in the present setup this tool is beyond the scope of this paper.

2) The hydrological discharge model or method presented in this paper doesn't require any modifications to the actual code of JSBACH itself but works by generating new input files for JSBACH every time the model is stopped and restarted (every 10 model years). As mentioned with the existing HD model of JSBACH there is already a lake model but this is unsuitable for modelling lakes whose size and depth changes or which cover

multiple grid boxes; a lake cell in this existing model is in effect a single reservoir with an infinite holding capacity and a set average residency time for water – this is effective for modelling the buffering effect of modern day lakes on hydrological discharge but not for modelling lakes with dynamically changing sizes. To introduce a new lake model will require the HD model within JSBACH to be modified; like many GCM components JSBACH was not design to cope with changing input data and thus this modification is likely to require a significant technical development within the model.

We agree that erosion of lake sills has a potentially important impact on paleo-hydrology (as we have now noted in the manuscript) and hope to address this in the lake model we are now developing.

**Changes to Manuscript:** We discuss the reason for omitting and consequences of omitting lakes in P20 L6-19 in the limitations section and provide discussion of the lack of sill erosion in P20 L20 – 27, also in the limitation section.

**2. Comment:** Fisher, T.G., 2005. Strandline analysis in the southern basin of glacial Lake Agassiz Minnesota and North and South Dakota, USA. Geological Society of America Bulletin 117 (11/12), 1481–1496.)

GIA and determines geographic and geoidal deflections and therefore errors from common simplified GIA schemes can also potentially significantly affect drainage routing. Again, all that is needed is a short statement to this effect. I note the authors do mentions "inaccuracies in the underlying topography" as an issue, but I think it would be useful to readers to spell out the key dynamical sources of this.

**Response:** Hypothetically our method can be used with any GIA scheme. Within PalMod it will be used with the VILMA model. We will add a short statement noting the GIA scheme used is a potentially important source of error and discuss the sources of this; both those in the original topography as a representation of the present day and those due to the imperfect transformation of that topography to represent a past time including accounting for erosion.

**Changes to Manuscript:** We have added discussion of errors related to the GIA scheme in the limitation section between P20 L34 and P21 L6.

**3. Comment:** My only other comment (which may be due to my ignorance of conda) relates to all the listed version specific required libraries in the code archive dynamic_hd_env.txt. I suspect this was auto generated and I would strongly urge the authors to reduce this to the bare bones required. Ideally, code should port without special language features, and the only versions issues should be to avoid known bugs in specific versions.

**Response:** Although this list was indeed automatically generated all of these libraries are required directly or indirectly by some element of the either the dynamic hydrological discharge code or the plotting code that goes with it. Extensive use of external libraries is typical for scientific python code. Many are included in the very extensive python standard library but some are part of a number of third party libraries. Python itself has a number of dependencies on other packages: openssl,readline,sqlite,tk,zlib,pip,setuptools and wheel. Scipy (which Numpy is part of; see http://www.numpy.org) is de facto the standard toolkit for scientific computing with Python. netCDF4-python is a package that allows netCDF4 files to be loaded and written in python. It has a lot of dependencies on other packages, namely: hdf5, libnetcdf, numpy, gfortran, zlib, curl, mkl, krb5 and openBLAS. Matplotlib is a very popular plotting package for Python (designed to emulate MATLAB's functionality) and also has a number of dependencies, namely: cycler, freetype, libpng, numpy, pyparsing, pyqt, python-dateutil, pytz, six, qt, sip and openssl. Also used for plotting is the backend cairo, on which depends fontconfig, libxml2, pixman and pycairo. We could create separate environments for the plotting code and reduce this list of packages somewhat however it would still be long enough with the netCDF4 module and its dependencies to require the use of a package manager and with the Conda package manager using the Anaconda repository the setup of these libraries should be done automatically. The Conda package manager and the associated Anaconda repository is used as it is a recommended method of installing Matplotlib according to Matplotlib's own install instructions and the top listed method of installing Scipy on the Scipy website. Its use allows close control over the python environment setup to ensure the correct version of each package is installed.

**Changes to Manuscript:** We have created a new release of the code (and updated the doi to it in the manuscript), see p22 L7 (latex-diff doesn't mark this change). This contains a set of instructions (as a tex file Dynamic_HD_Running_Instructions.tex in Dynamic_HD_Code/Dynamic_HD_Documentation/Dynamic_HD_Running_Instructions; this can be compiled with a latex compiler) with advice for how to setup the necessary environment to run the code without using conda alongside general guidance on how to run the code. The code now also has a flag to disable the use of conda and assume the user has taken care of setting up the necessary environment (details of which are given in the instructions).

**4. Comment:** # I'm confused here. You describe this as "present-day", so presumably just a routing matrix for present-day topography, but then below you indicate this does down-slope routing, in which case this does not need the "present-day" qualifier.

**Response:** Prior to this work on dynamic hydrology we had a fully static hydrology in the HD model in JSBACH with a set of prescribed river directions for the present day – this is still the HD model used for non-paleo work with JSBACH. The river directions for this were derived manually in several steps. First the orography was pit-filled; then a downslope routing was generated. The river directions thus generated were then themselves corrected by hand to ensure the correct paths for the world's main rivers and further corrections to the river directions were made based on a careful comparison of the catchment of each major river to reference catchments to ensure that each catchment was as far as possible correct.

**Changes to Manuscript:** We have added a sentence explaining how the manually corrected HD river directions were calculated in the introduction p2 L33-35.

**5. Comment:** # shouldn't this be run continuously in async mode? Ice sheet margins can significantly change drainage routes within a century.

**Response:** We will most likely run with a 10-year interval. True continuous running would be technically very difficult to implement within the MPI-ESM structure and this method is not fast enough to be run in such a way. It would be possible to run this method in parallel to provide a half yearly update to the river routing with a half year delay/offset however this would also be technically problematic as we would either have to stop and start the main ESM model every half year or allow it to let the river directions change during the course of a run. The former would likely be computationally inefficient and the latter very complicated to implement. Also, even if we implemented it would only be possible to update the ice margins every half year unless we also ran the GIA model VILMA as well which would be a further computational expense.

**Changes to Manuscript:** We have added an explanation of how (in PalMod at least) transient coupled ice-sheet-atmosphere-land-ocean paleo-simulations are likely to be run and given some justification of running every 10 years in the introduction section in p4 L27 – p5 L5.

**6. Comment:** "During the last glacial cycle, the courses of rivers in North America, .."

**should provide appropriate references**

**Response:** Agreed

**Changes to Manuscript:** We have added references to Teller 1990, Licciardi et al 1999, Mangerud et al 2004 and Wickert 2016 on P1 L20 in the introduction section.

**7. Comment:** "however such lakes are switched off entirely in the version of the HD model used for dynamic hydrological discharge modelling by this paper"

**why? Can the not be run with the dynamic upscaled topography?**

**Response:** The lakes provided in the existing HD model of JSBACH are not suitable for a changing topography; they can only be single grid cell in size (multiple lake grid cells can be placed together but they are independent

of one another) and their effect is to buffer throughflow. This gives a good representation of non-endorheic lakes for the present day but basin-filling lakes covering multiple grid cells will need a completely new model. As noted above this is future work and lies beyond the scope of this paper. The current lake models buffering effect was not seen as useful for paleolakes as we would have had to use present day parameters which would not be accurate for paleohydrologies. Given that the exact timing of discharge is not of great importance for the kind of coupled climate model paleo simulations this model is intended for it was thus preferred to drop the existing lake model entirely.

**Changes to Manuscript:** We note the existing present-day lake scheme is unsuitable for dynamic lakes on p2 L19-L20 in the introduction section.

**8. Comment:** # For ice sheet modelling, the meridional grid convergence towards the poles means that it makes much more sense to use eg 0.5 longitude by 0.25 latitude degree resolution.

**Response:** We need to use a 0.5-degree grid to match that of the existing Hydrological Discharge model code in JSBACH3. Implementing a 0.5 by 0.25-degree resolution would require significant technical changes to the JSBACH3 code itself; something this method has so far avoided as experience shows that changes to the land surface model itself are often time consuming to implement even when apparently simple in theory. Also, the generation of the present-day model orography and river directions on the 0.5-degree grid has required a significant amount of time, as many corrections are necessary to derived a realistic and consistent orography from an existing orography dataset (see our response to comment 4 and Hagemann and Dümenil 1998).

**Changes to Manuscript:** None

**9. Comment:** "We define the effective hydrological height of a cell within a DEM as the height of the highest point in the 'most likely' river"

**more succinct description would be "elevation of river sill within the cell" C3**

**Response:** Agreed, we will change to this wording.

**Changes to Manuscript:** We have added this wording on p8 L27 (in the DEM corrections section).

**10. Comment:** "Upscaling effective hydrological heights could also potentially be applied beneficially to Eurasia however we decided against doing so because of the significant additional effort required."

**I'm confused here as I understood from the reading that the core upscaling process is automated, in which case the effort is negligible. Farther down I note**

**Response:** Yes, it is automated… but see our next comment below.

**Changes to Manuscript:** None (but see below).

**11. Comment:** "The orography upscaling process (which need only be run once) takes approximately 25 minutes to run for the entire globe"

**so why not use the generated global field?**

**Response:** Despite notably improving the accuracy of the river directions produced the orography upscaling method doesn't remove all errors in present day river directions and some manual corrections are still required and it wasn't possible to make these for the entire globe due to time pressures. It is highly likely we will revisit this topic in future work on lakes and expand the use of this technique to the entire globe; possibly starting from an even higher resolution orography to reduce the number of manual corrections required.

**Changes to Manuscript:** We add an explanation of this on P12 L24-27 in the DEM corrections section.

**12. Comment:** "When these corrections are applied to an orography for a time other than the present day any relative corrections that are beneath ice sheets are temporarily suppressed until the region becomes ice free once more; thus the original unmodified height is always used for ice sheets"

**How do you handle the transition? Eg, say 200 m thick ice in the Rocky Mountains will not over-ride the existing topography?**

**Response:** We don't apply any special handling of the transition; we simply add the height of the ice-sheet to the unmodified 10-minute orography when deriving the orography of ice covered regions. We acknowledge this could be a source of error but it is not clear to the authors how to realistically model the growth of ice over the kind of narrow valley not resolved in the unmodified 10-minute orography and at what point an ice sheet growing over it would essentially smooth out this feature. This could potentially be future work although the first step would be to investigate how likely this is to occur and if so what effect it might have. It would likely be only a small impact unless such a thin ice sheet occurred for an extended period at a point critical for the determination of drainage directions.

**Changes to Manuscript:** We note this as a limitation in p21 L8-L15.

**13. Comment:** Another important limitation is the lack of verification for time-slices other than the present day (due to the lack of easily comparable data);

**This is an arguable statement. One could take the LGM topographic deflection from GIA, apply it to the hydro-1k DEM, overwrite ice topography, and then use eg the R hydrological flow solver (or equivalent in other GIS apps) to extract drainage maps and compare to your results.**

**Response:** We attempted to make this comparison but we were unable to remap the Hydro1k from its native Lambert Azimuthal Equal Area projection to a regular latitude-longitude projection with the tools available to us (unfortunately we don't have ARCGIS; QGIS wasn't able to handle the Hydro1k files and our own software for remapping was too slow to be of use). Also, the remapping process would likely have affected the hydrological tuning and would have thus necessitated a revalidation of the river directions produced. Instead we produced river directions using the 30 second resolution SRTM30 PLUS DEM in place of the Hydro1k for the LGM (and also the present day for validation) using the method suggested here. We didn't see any significant differences between the catchments produced thus and those produced by our method with the exception of a loss of a portion of its catchment by the Yukon due to differences in the flow directions on the ice-sheet itself. It is not clear that the version produced by the method suggested here is necessarily the correct one in this case; further discussion is given in the paper.

This process is also really a model rather than data; thus, this is a model to model comparison. We hope to make more extensive comparisons to data during future work on lakes; this will obviously be directed mostly at the lakes but necessarily validate also river directions.

**Changes to Manuscript:** We have removed "(due to the lack of easily comparable data);" from p20 L28-29. We present the results of the study suggested here but using the SRTM30 PLUS DEM instead of the Hydro1k DEM in the Application to an LGM simulation section on p18 L3-L21.

**14. Comment:** algorithmic pseudocode

**I found this hard to follow. eg it's not clear what all the variables such as p, d, h, i represent. As long as all the code is provided, I would suggest simplifying the pseudo code somewhat and use more descriptive names than single letter indices. Try running the pseudo code by some colleagues and see what doesn't make sense to them.**

**Response:** As suggested we have changed the single letter indices for more descriptive names. The pseudo-code is certainly difficult to understand however we are very hesitant to simplify the pseudo-code as it would no longer provide an essentially complete description of the algorithm and this would then make it very similar to the description provided in the main text. We have actually provided 4 different descriptions of the same

algorithm (if we count the code) and it is our intention that each one describes it to a different level of detail and thus with a different level of readability. We would not expect a reader to be able to understand the pseudo-code from a single reading; this is the purpose of the written description of the algorithm in the main text/appendix A (which is admittedly also complicated but is simplified as far as is possible); instead it functions as a template for anyone trying to rewrite the code in another language and as a reference to check details against while reading the description in the main text. The code itself (mostly found in fill_sinks.cpp and sink_filling_algorithm.cpp) is very considerably less readable still (despite being documented with comments in the proscribed C++ style), particularly as it shares a code base with two variants of the sink filling/river carving algorithm, has additional options for dealing with true sinks not described in the paper (and not presently used), is designed to be sufficiently abstract to be applicable to any arbitrary grid and as a consequence of these requirements uses the C++ features multiple-inheritance and generic programming (i.e. templates). We hope in the pseudo-code to present a description of the complete algorithm in a 'clean' code-like form without all of this additional technical infrastructure and without any account for issues such as memory management.

**Changes to Manuscript:** We have changed the names of most variables to be short words or abbreviations that reflect their purpose. (The pseudo-code is now located in Appendix B in response to comments by the other referee.) In the main text, p25 L11-L22, of the appendix we have extended the explanation of the meaning of the simple syntax used in the pseudo-code. We believe that combined with the description in appendix A, the altered variable names and improve description of the syntax, the pseudo-code should now be more understandable (although admittedly still too complex to enable it to be understood in a single read-through – I think it would require taking time to work through it slowly in combination with the description in the text to understand it properly.)

**15. Comment:** # I tried to set up the code set, but the required conda environment manager is not an available (apt-get accessible) package for Linux and installing this along with what I take are a lot of specific versions of python libraries.. was more than I was willing to go.

**Response:** For reasons unknown to the authors the Anaconda python distribution (of which the open source Conda is the package manager) can't be installed via apt-get. Linux installation instructions are given here:

https://conda.io/docs/user-guide/install/linux.html

**Changes to Manuscript:** We have created a new release of the code (and updated the doi pointing to it in the manuscript), see p22 L7 (not marked by latex-diff). This contains a set of instructions (as a tex file Dynamic_HD_Running_Instructions.tex in Dynamic_HD_Code/Dynamic_HD_Documentation/Dynamic_HD_Running_Instructions; this can be compiled with a latex compiler) with advice for how to setup the necessary environment to run the code without using conda, alongside general guidance on how to run the code. The code now also has a flag to disable the use of conda and assume the user has taken care of setting up the necessary environment (details of which are given in the instructions).

**16. Comment:** Figure 1,

**what does upscale by "meaning" mean?**

**Response:** We mean upscaling by taking the mean value of the height of all the fine DEM cells that are contained within the area covered by a given coarse DEM cell as the average height of that coarse DEM cell.

**Changes to Manuscript:** We have added this explanation to the caption of figure 1 on page 31.

**17. Comment:** Figure 8,

**would be much more informative to show change in volume flux eg in deci Sverdrups than change in water velocities.**

**Response:** We looked into this but decided against changing this plot. To convert to deci Sverdrups (or any unit with dimensions length cubed over time) we would have to multiply the value of each cell by its area. This would indeed make ballpark estimates of the change in river outflows from individual rivers much easier but would also mean that the value of each cell would be dependent on grid-box size and on this ocean grid grid-box size varies markedly. So, although changing to Sverdrups would have benefits we have decided not to do so as it would contravene the principles of good plot design and could be misleading. The proper way to compare the outflows of rivers in a meaningful way is either on a basin by basin basis (as we have done in figure 9) or a river by river basis (we have omitted this for reasons of space). Instead we have changed the unit from m/s to mm/day – the field we are plotting is actually P – E over the ocean with the river runoff added to P in the cells adjacent to a river mouth and mm/day is perhaps a more familiar measure of what is equivalent to a rainfall rate than m/s. This plot is best taken in combination with figure 9: figure 8 giving a visual guide to which rivers have changed; figure 9 being better for the extraction of numbers. We hope to look into more sophisticated ways of comparing changes in river outflows in future work.

**Changes to Manuscript:** We have changed figure 8 to mm/day on p37.

**18. Comment:** # small typos: ###
Artic Coast -> Arctic coast
Coast -> coast
northern Pacific -> Northern Pacific

**Response:** We will correct these in the final text.

**Changes to Manuscript:** We have added in these corrections on p19 L8-9.

**Response to Comments of Dai Yamazaki**

We thank Dai Yamazaki for his helpful comments.

**Comment 1:** Provide a literature review on the method for generating paleo river direction map. There is no citation to the previous method in the manuscript. If there is no previous study, please state so.

**Response:** We will add a literature review to the final text.

**Changes to Manuscript:** We have added a literature review in the introduction on p3 L29 – p4 L19.

**Comment 2:** The method section is difficult to understand. The authors stated that the method is applied for every 10-100 years, but the method description suggested some manual processing is needed. It is not clear how the authors could update the river direction dynamically during the paleo climate simulation.

**Response:** No manual processing is required during a run; however, a certain amount of manual processing was required to setup the orography corrections for use with this method.

**Changes to Manuscript:** We add clarification of how the river generation process will fit into a wider coupled transient ESM paleoclimate simulation in p4 L27-p5 L5 in the introduction section; clarify that the process is automatic during the run but the one-off development of a set of orography corrections is carried out by hand (albeit using automated tools) in p5 L27 – L31 in the methods overview section and further reinforce this point on p7 L10-13 in the DEM corrections section.

**Comment 3:** Related to above, only one river direction for the paleo climate was shown, though the method must have been applied multiple times during the simulation. As long as the authors stated that the method is for "dynamic river direction", the time series of gradual river direction change should be shown as a figure.

**Response:** We will add a short time series of plots showing the changing river routes in North America during deglaciation at around 12000 years BP. This should display a clear series of changes in the river routing.

**Changes to Manuscript:** We have added plots of the rivers in North America for 4 times during deglaciation in figure 11 on p40 and a brief section on this in the main text (section 5) p19 L23 – L27.

**Comment 4:** P2. L26: Negative values of are set to a constant. Please clarify which value was used. I guess, zero slope is also problematic, this the authors actually used "minimum threshold". Please clarify.

**Response:** The constant value used to replace negative values is 0.00001315. This value is also used to replace zero slope. Very small positive slopes are also replaced by a constant below a certain height difference threshold but the threshold is so small (a change in height of less than 0.00000000488281m) that this only acts as a guard against floating point errors and is not intended to affect any plausibly measurable height difference.

**Changes to Manuscript:** We have added this value to the text and noted that it also applies to zero slope in p2 L27-L28 in the introduction section.

**Comment 5:** P3. L4: The challenge for a paleoclimate simulation is to develop a method for periodically updating the river directions and flow parameters used with sufficient accuracy. Please provide reference to previous papers. How LGM river map was prepared in past studies. Is there any previous method which can treat "dynamic" river map generation?

**Response:** Most previous ESM based simulations of the last glacial cycle have used the technique of extending present day river directions to the sea (this was the suggestion for PMIP-3). Previous paleoclimate modelling in our own ESM model MPI-ESM has used this technique, see Ziemen et. al. (2014). A number of authors have tackled the problem of modelling river routing during the last glacial cycle. Wickert (2016) provides river maps for various time points during the deglaciation derived directly from a 30-second orography combined with various ice-sheet reconstructions. However, this technique would be too computationally expensive to run fully automatically every 10 years during a transient simulation. Tarasov and Peltier (2006) present a dynamic river routing and lake model for North America during the Younger Dryas that is in many ways similar to that presented here and from which the basic principle of upscaling of effective hydrological heights was taken. However, our new model uses a different combination of upscaling techniques and orography corrections from those of Tarasov and Peltier (2006) as well as a different grid.

Most previous simulations of the last glacial cycle that use coupled Global Circulation Models (GCMs) have only treated time-slices; transient simulations having usually been run only in models of intermediate complexity (EMICs). The first transient synchronously coupled GCM simulation of the deglaciation was Liu et. al. (2009). This used a time-varying prescribed forcing to simulate the release of glacial meltwater from rivers. However; the PalMod project, which the approach presented here is intended for, aims to run simulations that limit external forcings to solar and volcanic forcings, thus running transient models using a fully self-consistent earth system model and clearly precluding a proscribed forcing-based approach to meltwater runoff.

 In Ziemen et al. (subm) a simplified method was used, following similar ideas as we used here.

Liu, Z., Otto-Bliesner, B.L., He, F., Brady, E.C., Tomas, R., Clark, P.U., Carlson, A.E., Lynch-Stieglitz, J., Curry, W., Brook, E., Erickson, D., Jacob, R., Kutzbach, J., Cheng, J.: Transient simulation of last deglaciation with a new mechanism for bolling-allerod warming, Science, 325 (5938), pp. 310-314 , 2009.

Wickert, A. D.: Reconstruction of North American drainage basins and river discharge since the Last Glacial Maximum, Earth Surf. Dynam., 4, 831-869, https://doi.org/10.5194/esurf-4-831-2016, 2016.

Ziemen, F. A., Rodehacke, C. B., and Mikolajewicz, U.: Coupled ice sheet–climate modeling under glacial and pre-industrial boundary conditions, Clim. Past, 10, 1817-1836, https://doi.org/10.5194/cp-10-1817-2014, 2014.

**Ziemen, F.**, **Kapsch, M.-L.**, **Klockmann, M.** & **Mikolajewicz, U.** (submitted). Heinrich events show two-stage climate response in transient glacial simulations, Climate of the Past, in open review, doi: 10.5194/cp-2018-16. doi:10.5194/cp-2018-16

The other reference (Tarasov and Peltier, 2006) is already in the bibliography of the paper.

**Changes to Manuscript:** We have added this literature review to the text in the introduction section, p3 L29 – p4 L19.

**Comment 6:** P3. L20: False sinks also appear at higher resolutions due to various imperfections in the measurement of orography satellite: Recommended citation to the errors in DEM is: Yamazaki D., D. Ikeshima, R. Tawatari, T. Yamaguchi, F. O'Loughlin, J.C. Neal, C.C. Sampson, S. Kanae & P.D. Bates A high accuracy map of global terrain elevations Geo- physical Research Letters, vol.44, pp.5844-5853, 2017 doi: 10.1002/2017GL072874

**Response:** We will add this to the revised text.

**Changes to Manuscript:** We have added this reference to the text on p3 L25 of the introduction section.

**Comment 7:** P4:L8 A brief outline is given here From this section, it seems all steps are automated. While in Section 2.3 the authors mentioned the "by-hand" method which must be not automated. Please clarify this discrepancy. The "by-hand" correction was applied only once at the first step? Then, the description in Section 2.1 should be revised to avoid confusion.

**Response:** Yes, all the steps listed in Section 2.1 are automatic. The only manual work is the preparation of a set of orography corrections that is done offline beforehand and provided as an input file to the dynamic hydrology scripts at the start of a long transient run.

**Changes to Manuscript:** We have clarified this in the text in p5 L27- L31 in the methods overview section and reinforced this point further in P7 L10 – L13 in the DEM corrections section.

**Comment 8:** P4. L19: any intervening cells It is not clear that what "intervening cell" means.

**Response:** We mean a river passing through other cells on route to the cell in question. We will change 'i.e. without passing through any intervening cells' to 'i.e. without passing through any other cells first' and 'passing through intervening cells' to 'i.e. passing through other cells first' to make this clearer.

**Changes to Manuscript:** We have changed this in the text in the methods overview section p5 L34 – p6 L1.

**Comment 9:** P4. L24: 2.2 Changing the present-day base orography This section is very difficult to understand. What is the difference between the present-day-base-orography and present-day-reference-orography? Do they have different characteristics? Which data is used as "present-day-base-orography"? Do the two DEMs have the spatial resolution? Please clarify.

**Response:** As we intend to use our dynamic hydrology model both DEMs have the same spatial resolution (however, it would be theoretically possible to use a lower resolution present day base orography). The present-day base orography is the orography to which isostatic corrections (note these isostatic corrections are completely distinct and entirely independent of the other corrections, hydrological corrections, discussed here and throughout the paper) derived from a solid earth model (such as VILMA) will be applied during a transient paleoclimate simulation to generate the (general purpose 'physical') orography for a given past time. The present-day reference orography is the orography we used as a basis when developing the set of relative corrections used in our dynamic hydrology model; to be specific the present-day orography used by the ice sheet reconstruction ICE5G.

Our original plan was to apply the orography corrections (developed using the present-day reference orography) directly to the present-day base orography (which for the purposes of testing was ICE6G) and orographies derived from it; however, testing showed that this produced very poor results for the present-day: many previously corrected river directions were now wrong. Investigation showed that the ICE5G and ICE6G present-day orographies differed and that the poor results were due to these differences. Our solution is to essentially convert the orography used at a given time to be the changes due to isostatic corrections (generated by a solid earth model) applied to the present-day reference orography. This can't be done directly as solid earth model output is shared by several other components of the transient ESM setup so instead we convert the isostatic rebound corrections to relative isostatic corrections by subtracting the present-day base orography from the orography including the isostatic corrections. We then add these relative isostatic

corrections to the present-day reference orography and then add the (hydrological) corrections to give the working orography for a given past time for determining river routing.

**Changes to Manuscript:** We have rewritten the changing the present-day base orography section, p6 L10 – p7 L3 and also added clarification in the DEM corrections section, p12 L30 – p13 L5.

**Comment 10:** P5. L15: effective hydrological heights It is not clear what is the "effective hydrological height". Please write a brief explanation when it first appears.

**Response:** This is merely a list enumerating the names of the three techniques to be discussed below in detail. Effective hydrological height is defined on p6 L23. We feel that inserting the definition on P5 L15 would break the flow of the text and result in the definition being isolated from the main discussion of this topic.

**Changes to Manuscript:** We've added "Each of these techniques is described (including definitions of the terms 'intelligent river burning' and 'effective hydrological heights') individually below" to make it clear a definition is to come in the DEM section, p7 L16 – L 17

**Comment 11:** P6. L2: below a given threshold What threshold was used? Please clarify.

**Response:** A different threshold can be chosen for each region where intelligent burning is applied. The thresholds are chosen by hand in order to ensure that only the cells in the super fine orography through which the main river flows (in the region where the intelligent burning is being applied) are left unmasked. This is discussed on P6 L10.

**Changes to Manuscript:** We have added this explanation of how the thresholds are chosen to the text, p8 L14-L15 in the DEM section.

**Comment 12:** P6. L23: as the height of the highest point in the most likely river pathway It is not clear why "highest point" was used. Please explain.

**Response:** The highest point is used as this represents either the height of the (potential) river's bed as it flows into the cell (if the highest point on the path is on an edge; which it often will be) or the height of a lake sill if the cell is the boundary between an enclosed lake basin and a river basin; i.e. the height of a barrier blocking this pathway. Either way this is the height that will control whether a river flows along this pathway or another pathway.

**Changes to Manuscript:** Hopefully this point is now clear after we changed to using the river sill within the coarse cell as being the definition of effective hydrological height, p8 L27 in the DEM corrections section.

**Comment 13:** P6: L25: lowest highest point This expression is confusing. Please revise.

**Response:** We will revise the wording of this sentence to remove this.

**Changes to Manuscript:** We have reworded this sentence to remove this, p8 L29 – L31.

**Comment 14:** P7. L1: "flooding each coarse cell" It is not clear what "flooding" means here. Please explain in detail.

**Response:** Flooding is a term taken from the terminology of priority flood algorithms (upon which the structure of this algorithm is based). It is a quasi-physical description of the way the algorithm proceeds to process the fine cells that fit into the area of a given coarse cell; the processing order of the fine cells is the order in which they would fill with water were the entire coarse cell to be gradually filled with water starting from the lowest point on the coarse cell's boundary (and assuming that the coarse cell was surrounded by a continuous body of water such that disconnected basins within the cell would start flooding from separate edges).

**Changes to Manuscript:** We have added a brief version of this explanation to the text, p9 L8 – L12 in the DEM corrections section.

**Comment 15:** P7. L13: If the detailed description here is not needed to explain the main work (re-generating river maps during the climate simulation) proposed in the manuscript, I recommend to move this section to Appendix.

**Response:** Agreed, we will move the detailed description of the algorithm to the appendix.

**Changes to Manuscript:** We have moved the detailed description to Appendix A.

**Comment 16:** P10. L8: For this paper we upscale the unconditioned 30-second orography SRTM30 PLUS (Becker et al., 2009) to a 10-minute grid Is this used as "present day base orography" in Eq(3)? It is not clear how this data is used.

**Response:** The orography upscaling algorithm was applied to this and then the section for North America was extracted. This was converted to a set of relative corrections by subtracting the original "present day reference orography" and then added to the other relative corrections as described in the paper to form an improved set of relative corrections.

**Changes to Manuscript:** We have added clarification of this on p12 L18- L19 and also p12 L16, (both in the DEM section).

**Comment 17:** P13. L19: as all true sinks will be removed for actual paleoclimate simulation Is this a reasonable assumption? Please discuss.

**Response:** Water conservation is important for ESM based climate simulations and in particular for long transient simulations. In the case of paleo-climate simulations full treatment of true sinks requires a dynamic lake model so a terminal lake can be formed and filled until it either overflows (and thus ceases to be a sink and becomes a lake with through flowing water) or achieves a balance between inflow and evaporation. We are now actively developing such a lake model but that is future work beyond the scope of this paper. In the absence of such a dynamic lake model it is necessary to ensure water flowing to true sink points still reaches the sea by other means.

The standard (non-dynamic) HD model in JSBACH which predates this work and which this work is based upon conserves water by adding any water reaching true sink points to the outflow from rivers into the sea, distributing that water in proportion to the size of those rivers.

Here we choose to use an alternative scheme and by removing all true sinks force the water flowing to true sinks into an adjacent river basin across the lowest point along the watershed of the true sinks basin. Changing to this alternative scheme increased the total water flux into the Indo-Pacific and reduced that into the Atlantic for the present day as noted in the paper; however, this had little effect on large-scale ocean circulation. Changing to this alternative scheme will facilitate future work on lakes and ensure the water that would fill any large lakes during deglaciation is routed via the correct spill pathway without actually modelling the lake itself. The removal of true sinks is a necessary assumption in the absence of dynamic lakes. We acknowledge the lack of dynamic lakes will be a significant error in this dynamic hydrology scheme; however, we should be able to generate useful scientific results without them (we will just lack the buffering effect of filling periglacial lakes and the periglacial lake outburst 'mega'-floods) and given the timeframe of PalMod we are likely to make significant use of the current scheme in scientific runs before it is possible to add in dynamic lakes.

**Changes to Manuscript:** Alongside the discussion of the omission of lakes added to the limitations section, P20 L6-L19, we have also justified this omission of true sinks in p13 L31- L34 in the false sink removal section.

**Comment 18:** P14. L14: in all cases they are either due to minor errors in the manually corrected JSBACH river directions The error could be "minor" for the purpose of climate simulation, but the same error could be critical for different purpose. The authors have to acknowledge the method is suitable for climate simulations, but the accuracy could be not adequate for different use (such as water resource assessment or flood risk modelling).

**Response:** The manually corrected JSBACH river directions predate this work on dynamic hydrology by many years and are an essential component of the Hydrological Discharge model either as part of JSBACH or run as a

standalone model of present day global hydrology. They have thus been used for a variety of non-paleoclimate modelling purposes for which they have been deemed appropriate either as part of a standalone HD model, a standalone land-surface model, a standalone global hydrology model, or as part of a full Earth System Model (a coupled ocean, atmosphere and land model). Note that the HD model as part of the global hydrology model MPI-HM (Stacke & Hagemann 2012) was not performing noticeably different than other global hydrology models (Haddeland et al. 2012). In this paper our focus is exclusively on making this present-day HD model into a dynamic HD model for paleoclimate modelling and it is not our intention to comment on the suitability of the base HD model for other purposes; it is described and discussed in the references given in the paper (Hagemann and Dümenil,1998b; Hagemann and Dümenil Gates, 2001).

**Haddeland, I., D.B. Clark, W. Franssen, F. Ludwig, F. Voß, N.W. Arnell, N. Bertrand, M. Best, S. Folwell, D. Gerten, S. Gomes, S.N. Gosling, S. Hagemann, N. Hanasaki, R. Harding, J. Heinke, P. Kabat, S. Koirala, T. Oki, J. Polcher, T. Stacke, P. Viterbo, G.P. Weedon and P. Yeh, 2011.** Multi-Model Estimate of the Global Terrestrial Water Balance: Setup and First Results . *J. Hydrometeor.* **12**, 10.1175/2011JHM1324.1, 869-884.

**Stacke, T. and S. Hagemann, 2012** Development and validation of a global dynamical wetlands extent scheme. *Hydrol. Earth Syst. Sci.* **16**, doi:10.5194/hess-16-2915-2012: 2915-2933

**Changes to Manuscript:** We have added a comment on the evaluation of the HD model for the present day and the two references given here to the introduction section, p3 L8-L10.

**Comment 19:** P14. L31: Application to a LGM simulation P14.L32: Figure 7 I guess, many readers want to check how the algorithm calculates gradual change in river directions following the change in orography. Given that the authors stated in Section 2.1 that "River directions are regenerated approximately every 10-100 years", there must me multiple river maps for glacier period simulations. However, the authors showed only 1 river map for the paleo climate simulation. I think this is not adequate to proof the usefulness of the proposed method. I here suggest to show the time series of river map development, by focusing on some locations where the authors can observe the gradual change of river directions during the paleo simulation. Otherwise, it is difficult to state that the proposed method is "dynamics" river direction mapping.

**Response:** As discussed above, we will add a short time series of plots showing the changing river routes in North America during deglaciation at around 12000 years BP. This should display a clear series of changes in the river routing.

**Changes to Manuscript:** We have added this in figure 11 on p40 and a brief section on this in the main text (section 5) p19 L23 – L27.

**Comment 20**: Figure 7: Please use different color for the "new land" or "new ocean" in Figure 7 to show the change in land sea mask.

**Response:** We will highlight new land with a distinct shade of brown and new ocean with a distinct shade of blue.

**Changes to Manuscript:** We have highlighted new land with a distinct shade of brown in figure 7 on p36 and noted this in the caption of the figure. We have not highlighted new sea as the quantity of new sea at the LGM is insignificant (sea level was considerably lower at the LGM).

[revised manuscript text omitted]

6:    Push *AnnotatedCell* onto *Open*

7:    *CatchmentNumbers*($C_F$) ← count

8:    count ← count + 1

9:    *PathInitialHeights*($C_F$) ← SEALEVEL

10:    *Closed*($C_F$) ← **true**

11: **end for**

12: **for all** fine cells $C_F$ on the edges of *FineDEMSection* **do**

13:    **if** *FineLandSeaMaskSection*($C_F$) is SEA **then**

14:       **skip to next iteration of loop**

15:    **end if**

16:    *AnnotatedCell* ← (  pos ← $C_F$, height ← *FineDEMSection*($C_F$), initialheight ← *FineDEMSection*($C_F$), edgeid ←TOP/BOTTOM/LEFT/RIGHTEDGEID(as appropriate), len ← 1, greatestperpdist ← 0, catchid ← count)

17:    Push *AnnotatedCell* onto *Open*

18:    *CatchmentNumbers*($C_F$) ← count

19:    count ← count + 1

20:    *PathInitialHeights*($C_F$) ← *FineDEMSection*($C_F$)

21: **end for**

22: **return** *Open*, *Closed*, *PathInitialHeights*, *CatchmentNumbers*
* * *
**Algorithm 3** Orography upscaling algorithm process queue

Compare to steps 2(c)-(e) of the description given in Appendix A.
* * *
**Require:** *FineDEMSection, FineLandSeaMaskSection, PathInitialHeights, Open, CatchmentNumbers, Closed, AnnotatedCell*, $C_C$, *Coarse-DEM*

1: **while** *Open* is not empty **do**

2:    $C_F$,  $height_{CEN}$, $initialheight_{CEN}$, $edgeid_{CEN}$, $len_{CEN}$, $greatestperpdist_{CEN}$, $catchid_{CEN}$ ← POP(*Open*)

3:     $greatestperpdist_{CEN}$ ← MAX($greatestperpdist_{CEN}$, current perpend

4:    *Closed*($C_F$) ← **true**

5:    **if** $C_F$ is an edge cell **or** $C_F$ neighbours one or more cells which are SEA in *FineLandSeaMaskSection* **then**

6:       **if** $len_{CEN}$ > MINIMUMPATHTHRESHOLD **then**

7:          **if** (**not** (identity of edge at $C_F$) = $edgeid_{CEN}$) **or**
             ($greatestperpdist_{CEN}$ > MINIMUMSEPARATIONFROMINITIALEDGETHRESHOLD **and** **not** $edgeid_{CEN}$ = RIVERMOUTH)
             **then**

8:             *CoarseDEM*( $C_C$) ← $height_{CEN}$

9:             **exit loop**

10:          **end if**

11:       **end if**

12:    **end if**

13:    **for all** neighbouring cells $N$ of $C_F$ **do**

14:       **if** *Closed*($N$) is **true then**

15:          **if** *PathInitialHeights*($N$) < $initialheight_{CEN}$ **or**
             *CatchmentNumbers*($N$) = $catchid_{CEN}$ **or**
             (*PathInitialHeights*($N$) = $initialheight_{CEN}$ **and** *CatchmentNumbers*($N$) > $catchid_{CEN}$) **or**
             *FineLandSeaMaskSection*($N$) is SEA)) **then**

16:             **skip to next iteration of loop**

17:          **end if**

18:       **end if**

19:        *FineDEMSection*($N$) ← MAX(*FineDEMSection*($N$), $height_{CEN}$)

20:       *Closed*($N$) ← **true**

21:       *CatchmentNumbers*($N$) ←  ← $catchid_{CEN}$

22:       *PathInitialHeights*($N$) ←  ← $initialheight_{CEN}$

23:       neighbour path length  $len_N$ ← ($len_{CEN}$ + 1) **if** $N$ is a non-diagonal neighbour of $C_F$ **else**  $len_N$ ← ($len_{CEN}$ + $\sqrt{2}$)

24:       *AnnotatedCell* ←  $pos$ ← $N$, $height$ ← *FineDEMSection*($N$), $initialheight$ ← $initialheight_{CEN}$, $edgeid$ ← $edgeid_{CEN}$, $len$ ← $len_N$, $greatestperpdist$ ← $greatestperpdist_{CEN}$, $catchid$ ← $catchid_{CEN}$)

25:       Push *AnnotatedCell* onto *Open*

26:    **end for**

27: **end while**

28: **return** *CoarseDEM*